

# Ice nucleating particles in the marine boundary layer in the Canadian Arctic during summer 2014

Victoria E. Irish[1], Sarah J. Hanna[1], Megan D. Willis[2], Swarup China[3], Jennie L. Thomas[4], Jeremy J. B. Wentzell[5], Ana Cirisan[6], Meng Si[1], W. Richard Leaitch[7], Jennifer G. Murphy[2], Jonathan P. D. Abbatt[2], Alexander Laskin[8], Eric Girard[6], Allan K. Bertram[1]

[1] Department of Chemistry, University of British Columbia, 2036 Main Mall, Vancouver BC, V6T 1Z1, Canada

[2] Department of Chemistry, University of Toronto, 80 St George Street, Toronto, Ontario, ON M5S 3H6, Canada

[3] Environmental Molecular Sciences Laboratory, Pacific Northwest National Laboratory, Richland, WA 99354, USA

[4] Laboratoire Atmosphères, Milieux et Observations Spatiales, Université Pierre et Marie Curie, 4 place Jussieu, 75252 Paris Cedex 05, France

[5] Air Quality Research Division, Environment and Climate Change Canada, 4905 Dufferin Street, Toronto ON, M3H 5T4, Canada

[6] Département des sciences de la Terre et de l'atmosphère, Université du Québec à Montréal, 201, avenue du Président-Kennedy, Montréal, Québec, QC H2X 3Y7, Canada

[7] Climate Research Division, Environment and Climate Change Canada, 4905 Dufferin Street, Toronto ON, M3H 5T4, Canada

[8] Department of Chemistry, Purdue University, West Lafayette, IN 47907, USA

*Correspondence to*: Allan Bertram (bertram@chem.ubc.ca)

**Abstract.** Ice nucleating particles (INPs) in the Arctic can influence climate and precipitation in the region; yet our understanding of the concentrations and sources of INPs in this region remain uncertain. In the following we 1) measured concentrations of INPs in the Canadian Arctic marine boundary layer during summer 2014 on board the CCGS *Amundsen*, 2) determined ratios of surface areas of mineral dust aerosol to sea spray aerosol, and 3) investigated the source region of the INPs using particle dispersion modelling. Average concentrations of INPs at -15, -20 and -25 °C were 0.005, 0.044, and 0.154 L$^{-1}$, respectively. These concentrations fall within the range of INP concentrations measured in other marine environments. For the samples investigated the ratio of mineral dust surface area to sea spray surface area ranged from 0.03 to 0.09. Based on these ratios and the ice active surface site densities of mineral dust and sea spray aerosol determined in previous laboratory studies, our results suggest that mineral dust is a more important contributor to the INP population than sea spray aerosol for the samples analysed. Based on particle dispersion modelling, the highest concentrations of INPs were often associated with lower latitude source regions such as the Hudson Bay area, eastern Greenland, or northwestern continental Canada. On the other hand, the lowest concentrations were often associated with regions further north of the sampling sites and over Baffin Bay. A weak correlation was observed between INP concentrations and the time the air mass spent over bare land, and a weak negative correlation was observed between INP concentrations and the time the air mass spent over ice and open water. These combined results suggest that mineral dust from local sources is an important contributor to the INP population in the Canadian Arctic marine boundary layer during summer 2014.





## 1 Introduction

Ice nucleating particles (INPs) initiate the heterogeneous formation of ice in clouds at temperatures warmer than required for homogeneous freezing. INPs are important since they can change the frequency and properties of ice and mixed-phase clouds. The frequency and properties of clouds in the Arctic have been shown to be especially sensitive to
concentrations of INPs, yet our understanding of the concentrations and sources of INPs in this region remain uncertain (Coluzza et al., 2017; Creamean et al., 2018; Harrington et al., 1999; Kanji et al., 2017; Korolev et al., 2017).

Examples of atmospherically relevant INPs include, but are not limited to, mineral dust particles and sea spray aerosol (DeMott et al., 2016; Després et al., 2012; Hoose and Möhler, 2012; Murray et al., 2012; Niemand et al., 2012; Szyrmer and Zawadzki, 1997). Sea spray aerosol is generated by a bubble bursting mechanism at the ocean surface
(Blanchard, 1964). Recent work has shown that the sea surface microlayer and bulk seawater contain INPs (Irish et al., 2017; Rosinski et al., 1986; Schnell, 1977; Schnell and Vali, 1975, 1976; Wilson et al., 2015). Modelling studies have also suggested that sea spray aerosol can be a significant contributor to the INP population in marine environments when the source of other INPs is small (Burrows et al., 2013; Vergara-Temprado et al., 2017; Wilson et al., 2015). However our understanding of the flux of INPs from the ocean to the atmosphere is incomplete, and more studies are needed to understand
when and where sea spray aerosol are a significant contributor to INP concentrations in the atmosphere.

Mineral dust is transported to the atmosphere by wind erosion, which is sensitive to factors like soil composition, soil moisture, and wind velocity (Ginoux et al., 2001). Mineral dust has been identified as an important contributor to the atmospheric INP population in many field and laboratory studies (Atkinson et al., 2013; Boose et al., 2016; Chen et al., 2018; Conen et al., 2011; Connolly et al., 2009; Creamean et al., 2013; DeMott et al., 2015; Eastwood et al., 2008; Hill et al.,
2016; Kaufmann et al., 2016; Klein et al., 2010; Murray et al., 2012; Niedermeier et al., 2010; Niemand et al., 2012; O'Sullivan et al., 2014; Prenni et al., 2009a, 2009b; Rangel-Alvarado et al., 2015; Steinke et al., 2016; Wex et al., 2014; Wheeler et al., 2015). Modelling studies have also confirmed that mineral dust particles are important atmospheric INPs (Alizadeh-Choobari et al., 2015; Atkinson et al., 2013; Burrows et al., 2013; Hendricks et al., 2011; Hoose et al., 2010a; Lohmann and Diehl, 2006; Prenni et al., 2009b; Vergara-Temprado et al., 2017).

When glaciers and permafrost in the Arctic melt erodible soil is exposed. The increased areas with erodible soil can be a potential source of mineral dust in the Arctic (Huang et al., 2015). The ice nucleation properties of dry mineral dust from Thule, Greenland were measured by Fenn and Weickmann (1959) and they found it could nucleate ice at temperatures as warm as -5 °C. Groot Zwaaftink et al. (2016) also suggested through a modelling study that during the summer local mineral dust sources in the Arctic (mineral dust from latitudes north of 60 ° N) can dominate the total mineral dust
concentrations at the surface. However, previous studies have not yet shown that mineral dust from regional erodible soil could be a major source of INPs in the atmosphere in the Arctic.

In the past 30 years, warming in the Arctic has decreased sea ice and land snow by approximately 20 % and 13 %, respectively (Derksen and Brown, 2012). This may have led to an increase in sea spray particles and mineral dust particles



from local sources in the region and, as a result, an increase in INPs. Because of the continuing warming trend in this region, the concentration of INPs from these local sources may continue to increase with important implications for the frequency and properties of ice and mixed-phase clouds as well as climate in the region. To evaluate the scale of this feedback mechanism, studies are needed to determine the concentrations and sources of INPs in the Arctic.

5    To help address the issues raised above we: 1) determined concentrations of INPs in the Canadian Arctic marine boundary layer during summer 2014, 2) measured the ratio of surface areas of mineral dust particles to sea spray particles, and 3) investigated the source region of the INPs using a particle dispersion model. The specific goals of this study were to quantify the concentrations of INPs in the Canadian Arctic marine boundary layer and to provide insights into the source of INPs in this region during the summer.

## 2 Experimental

### 2.1 Sampling locations

All measurements and sample collection were performed on board the CCGS *Amundsen* as part of the NETwork on Climate and Aerosols; addressing key uncertainties in Remote Canadian Environments (NETCARE). The 28 sampling locations are shown in Fig. 1, and the sampling dates, times, and coordinates are detailed in Table S1. The air temperature, relative humidity, and wind speed during sampling are shown in Fig. S1. INP concentrations from a subset of the locations (indicated with blue rings around the red symbols in Fig. 1) were previously reported in DeMott et al. (2016) but are also included here as they were collected during the same expedition and with the same methodology. The data reported in DeMott et al. (2016) only included sites in Baffin Bay, days where it did not rain and conditions when the apparent wind direction measured on the ship was between 0-90 degrees or 270-360 degrees, where 0/360 corresponds to the bow of ship (where the apparent wind direction is defined as the wind direction experienced by an observer on the moving ship as opposed to the true wind direction, which is defined as the wind direction experienced by an observer when the ship is stationary).

### 2.2 Quantifications of INPs

To determine the concentration of INPs, atmospheric particles were collected on hydrophobic glass slides using a micro-orifice single stage impactor (MOSSI; MSP corp., Shoreview, MN, USA). The freezing properties of the collected particles were then determined with the droplet freezing technique (DFT). The combination (MOSSI-DFT) is similar to the micro-orifice uniform deposit impactor droplet freezing technique (MOUDI-DFT) recently used to determine the size resolved concentrations of INPs (Mason et al., 2015b, 2015a, 2016). The main difference between the MOSSI-DFT technique and the MOUDI-DFT technique is the use of a single stage impactor compared to a multistage impactor. The use of a single stage impactor simplifies the analysis and reduces collection time but sacrifices size information. The MOSSI-





DFT technique is also similar to the technique used by others to measure deposition freezing (Knopf et al., 2010, 2014, Wang et al., 2012b, 2012a).

### 2.2.1 Micro-orifice single stage impactor

The MOSSI was located on the port side of the bridge on the ship, approximately 10 m in front of the ship's
smokestack. During sampling the flow rate through the MOSSI was 10 L min$^{-1}$, resulting in particles with aerodynamic diameters > 0.18 μm being collected on the hydrophobic glass slides placed within the MOSSI. The collection time of samples with the MOSSI for INP analysis was approximately 20 minutes. The MOSSI sampled air through a louvered total suspended particulate (TSP) inlet, which was approximately 15 m above sea level. The nozzle plate within the MOSSI contained 300 micro-orifices. As a result, particles collected on the hydrophobic glass slides beneath the nozzle plate were
concentrated into 300 spots. After collection the hydrophobic glass slides containing the particles were stored at 4 °C for no longer than 3 months prior to analysis.

Particle rebound can be an issue with an inertial impactor such as the MOSSI. Particle rebound occurs when particles impact the collection substrate but are not retained. Rebound has been shown to be reduced at RH values above 70 %, although this depends on the chemical composition of the particles (Bateman et al., 2013; Chen et al., 2011; Fang et al.,
1991; Lawson, 1980; Saukko et al., 2012; Vasiliou et al., 1999; Winkler, 1974). During collection with the MOSSI the RH was always well above 70 % (Figure S1) with an average of 93 %. Furthermore, field measurements of INP concentrations using the MOUDI-DFT (a method similar to the MOSSI-DFT) have shown good agreement with INP concentrations measured with an instrument that is not susceptible to particle rebound (a continuous flow diffusion chamber) when the RH of the sampled aerosol was as low as 40 % (DeMott et al., 2017; Mason et al., 2016). Nevertheless, particle rebound cannot
be ruled out, and therefore the INP concentrations reported here should be considered as lower limits.

### 2.2.2 Droplet freezing technique

The droplet freezing technique (Koop et al., 1998; Mason et al., 2015b) was used to determine the concentration of INPs in the immersion mode collected on hydrophobic glass slides using the MOSSI. Briefly, the hydrophobic glass slides containing the collected particles were located in a temperature and relative humidity controlled flow cell, coupled to an
optical microscope with a charged-coupled device camera and a 1.25x objective (Axiolab; Zeiss, Oberkochen, Germany). Typically between 15-25 spots of particles (out of the 300 spots generated by the micro-orifices in the nozzle plate) could be monitored in the CCD field of view, which was approximately 12.25 mm$^2$ in area. Water was then condensed on the hydrophobic glass slides by decreasing the temperature to 2 °C and flowing a gas (pure Helium) with a dew point of greater than 2 °C over the hydrophobic glass slides. This resulted in water droplets (with diameters between 100 to 500 μm)
condensing on the spots (referred to here as spot droplets) as well as water droplets condensing on other areas of the slides (referred to here as non-spot droplets). After the droplets were condensed, the temperature of the flow cell was decreased at a rate of 10 °C min$^{-1}$. From videos recorded while the temperature was decreased, the freezing temperature of each droplet was



manually determined by observing the change in the droplet's optical properties. The droplets that contained spots of deposited particles were also identified from these videos. For comparison purposes, hydrophobic glass slides that were not exposed to atmospheric particles were also processed in the same way as described above and labelled as blanks.

The number of INPs as a function of temperature, *#INP(T)*, was calculated for each experiment using the following
equation:

$$\# INP(T) = \left( -\ln\left( \frac{N_u(T)}{N_s} \right) N_s \right) \tag{1}$$

Where $N_u$ is the number of unfrozen droplets covering the spots, and $N_s$ is the number of spots in the field of view. Equation 1 accounts for the possibility that each droplet covering a spot can contain multiple INPs (Vali, 1971).

Equation 1 assumes that each spot was covered by only one droplet. For cases when more than one droplet formed
on a spot, the first droplet that froze was considered in Eq. 1. This was expected to give an equivalent result to the case of only one droplet condensing on the spot. For cases when one droplet contained two spots (this occurred for 2% of the total number of spots in all experiments), an upper limit to the number of INPs was calculated by assuming two droplets covered the two spots and both droplets froze at the observed freezing temperature. A lower limit was calculated by assuming the two spots were covered by two droplets with one droplet freezing at the observed freezing temperature, and the other droplet
freezing at -37 °C (approximately the homogeneous freezing temperature). If one droplet contained 3 or more spots a similar procedure to the above was used to calculate the upper and lower limits to *#INP(T)*.

Freezing of droplets that did not cover spots was a relatively rare occurrence at the temperature range we focused on in this manuscript (≥ -25 °C; see Section 3.1). For example the ratio of frozen non-spot droplets to frozen spot droplets was 0.02 and 0.07 at -25 °C and -20 °C, respectively. We assumed these relatively rare occurrences were due to particles < 0.18
μm in diameter that were not focused into spots or due to rebound of a small fraction of the particles off the hydrophobic glass slides. To take into account the INPs not concentrated into the spots, we added the number of frozen non-spot droplets at each temperature to Eq. 1. Since the freezing of non-spot droplets was a relatively rare occurrence, we did not apply the Vali correction (Vali, 1971) to these freezing events.

Approximately 2 % of the freezing events in our experiments occurred by contact freezing between -16.2 and -34.8
°C. Contact freezing occurred when a frozen droplet grew in size, due to the Wegener-Bergeron-Findeisen process (Findeisen, 1938), and caused the freezing of a neighbouring unfrozen droplet. When calculating concentrations of INPs, contact freezing events were excluded.

The atmospheric concentration of INPs as a function of temperature, [INP(T)], was calculated with the following equation:

$$[INP(T)] = \# INP(T) . \frac{300}{N_s} . \frac{1}{V}$$


(2)





Where $V$ is the volume of air sampled, and the ratio of $300/N_s$ takes into account that only a fraction of the total number of spots in the sample were observed in a freezing experiment.

**2.3 Effect of ship emissions on measured INP**

To determine if particles from the ship's smokestack affected the measured INP concentrations, we first
investigated the relationship between INP concentrations measured on the ship and the gas-phase HONO concentrations, a product of the reaction between $NO_x$ from the ship smokestack and water (Von Glasow et al., 2003). HONO was measured by a chemical ionisation mass spectrometer located on the bridge of the ship about 5 m in front of the smokestack. No correlation was observed between HONO and INP concentrations at -25 °C (R = 0.05, p = 0.403).

Second we separated our INP results into samples that were not exposed to smokestack emissions based on
measured wind direction and wind speed, and samples that may have been exposed to smokestack emissions based on measured wind direction and wind speed. When the apparent wind direction measured on the ship was between 0-90° and 270-360° (where 0°/360° = bow of ship) and when the apparent wind speed (minute average) was higher than 2.5 m s$^{-1}$ for the entire collection time, we assumed that the samples were not exposed to smokestack emissions. Within the uncertainty of the measurements, the INP concentrations measured when samples were not exposed to smokestack emissions (based on the
apparent wind direction and speed) are the same as INP concentrations measured when samples may have been exposed to smokestack emissions (Fig. S2). Since the criteria discussed above and the results from the HONO analysis do not suggest measured INP concentrations were influenced by the smoke stack emissions, all samples that were collected have been included in this study.

**2.4 Particle dispersion modelling**

FLEXPART-WRF (Brioude et al., 2013), a version of the Lagrangian particle dispersion model FLEXPART (Stohl et al., 2005), was used to investigate the potential emission source regions of the INPs. FLEXPART-WRF is driven by meteorology from the Weather Research and Forecasting (WRF) model (Skamarock et al., 2008), and was run in backward mode. The simulation domain for FLEXPART-WRF is shown in Fig. S3.

WRF 3.5.1 was run for the 2014 Amundsen campaign using initial and boundary conditions from the European
Centre for Medium-Range Weather Forecasts (ECMWF) operational analysis (a grid resolution on 0.25°). The ECMWF wind, temperature, and RH were used to nudge the WRF run every 6 hours above the atmospheric boundary layer. A full list of the parameterizations and options used for the WRF simulations is given in Table 1 of Wentworth et al. (2016).

FLEXPART-WRF was run in backward mode at 20 minute intervals along the ship track. For each run 100,000 particles were released from the ship's location in a volume of 100 x 100 m in the horizontal, and from 0 to 60 m above mean sea
level in the vertical. The particles were followed backward for seven days with output generated hourly.

As mentioned above, each INP sample was collected over a period of approximately 20 minutes. As a result, one FLEXPART-WRF run overlapped with each INP sampling period. The FLEXPART-WRF runs that overlapped in time with





the INP sampling periods were used to produce potential emission sensitivity (PES) plots for the INP samples. A PES plot was produced by integrating the FLEXPART-WRF output over the 7 days prior to the release of particles. The value of the PES in a particular grid cell is proportional to the particles residence time in that cell. Since this study is concerned with INP sources from the surface, only particles within the footprint layer (0 to 300 m altitude) are considered when calculating PES

values, and we report PES values for the footprint layer.

**2.5 Statistical analyses**

Pearson correlation analysis was used to compute correlation coefficients (R). P values were also calculated to determine if the correlations were statistically significant at the 95 % confidence level ($p < 0.05$).

**2.6 Computer controlled scanning electron microscopy with energy dispersive X-ray spectroscopy (CCSEM-EDX)**

Immediately following the collection of each sample for INP analysis, additional particle samples were collected to determine the ratio of mineral dust surface area to sea salt surface area by computer controlled scanning electron microscopy with energy dispersion X-ray spectroscopy (CCSEM-EDX). Particles were collected on transmission electron microscopy (TEM) grids (carbon 200 mesh; Ted Pella) using the same MOSSI used to collect INP samples. Collection time of samples for CCSEM-EDX was approximately 20 minutes. The samples were kept at room temperature for a maximum of 38 months

before analysis. Since a long collection time was used (20 minutes), particles in the spots directly below the micro-orifices of the nozzle plate in the MOSSI were too close together to identify individual mineral dust and sea salt particles using CCSEM-EDX. To overcome this issue, we only analysed particles on the edge of the spots directly below the micro-orifices of the nozzle plate with CCSEM-EDX.

Due to time constraints, only three samples (two with a high [INP(T)] and one with a low [INP(T)]) were analysed

by CCSEM-EDX to determine the ratios of mineral dust surface area to sea salt surface area. The method of using CCSEM-EDX to study atmospheric particles is described by Laskin et al. (2006). Particles with sizes between 0.15 to 5 µm (area equivalent diameters) were analysed. First, the atomic percentages of each particle were determined from EDX spectra. Then, based on the atomic percentages of each particle, particles were classified as sea salt, mineral dust, or other using the scheme shown in Fig. S4, which is based on the work by Laskin et al. (2012). After each particle was classified, the surface

areas of mineral dust particles and sea salt particles were determined using 2D projected images recorded by SEM. Note that the actual surface area of mineral dust is underestimated using this method due to surface irregularities and complex topology. The ratio of mineral dust surface area to sea salt surface area was then determined by dividing the surface area of mineral dust for each sample by the surface area of sea salt for the same sample.





## 3 Results and discussion

### 3.1 Measured INP concentrations

The measured concentrations of INPs, [INP(T)], sampled in the Arctic are shown in Figs. 2b and 2c. The measured [INP(T)] on hydrophobic glass slides that were not exposed to atmospheric particles, referred to as "blanks", are shown in
red in Figs. 2a and 2c. Freezing of the blanks occurred over the range of -25.9 °C to -38.4 °C. For the droplet sizes and cooling rates used here homogeneous freezing occurs at approximately -37 °C. Therefore the freezing that occurred in the blanks at temperatures above approximately -37 °C was due to heterogeneous freezing likely caused by the hydrophobic glass slides. In the following we will focus on [INP(T)] at temperatures of -25 °C and warmer since no freezing from the blanks was observed in this temperature range. A full time series of [INP(T)] at -15 °C, -20 °C, and -25 °C are reported in
Fig S5.

In Fig. 3 we compare recent measurements of [INP(T)] from several field campaigns in marine environments with the average concentrations measured in the current study. Figure 3 illustrates that the average INP concentrations measured in the current study fall within the range of INP concentrations measured in other marine environments. This observation, however, does not confirm that sea spray aerosol was the major source of INPs during the studies reported here.

### 15   3.2 Measured ratios of mineral dust surface area to sea salt surface area

For three samples (two with high [INP(T)] and one with low [INP(T)]), we calculated the ratios of mineral dust surface area to sea salt surface area using CCSEM-EDX. The two samples corresponding to high [INP(T)] were collected on July 21$^{st}$ and 25$^{th}$. The sample corresponding to a low [INP(T)] was collected on July 29$^{th}$. In Table S2 we report the total number of particles analysed by CCSEM-EDX for each sample, and the fraction of particles classified as mineral dust and
sea salt particles. Shown in Fig. 4a are the calculated ratios of mineral dust surface area to sea salt surface area using surface area measurements from CCSEM-EDX. For the three samples, this ratio ranged from 0.03 to 0.09. Using this ratio we estimated the ratio of [INP(T)] from mineral dust, [INP(T)]$_{MD}$, to [INP(T)] from sea spray, [INP(T)]$_{SS}$, using the following equation:

$$\frac{[INP(T)]_{MD}}{[INP(T)]_{SS}} = \frac{n_s(MD).S_{MD}}{n_s(SS).S_{SS}} \tag{3}$$

Where $n_s(SS)$ is the ice active surface site density for sea spray aerosol, $n_s(MD)$ is the ice active surface site density for mineral dust, and $S_{SS}$ and $S_{MD}$ are the total surface areas measured by CCSEM-EDX for sea salt and mineral dust, respectively. The $n_s(SS)$-values were determined using laboratory data from DeMott et al. (2016). The $n_s(MD)$-values were calculated using the exponential function reported by Niemand et al. (2012) that was determined from freezing data of Asian dust, Saharan dust, Canary Island dust, and Israel dust. For details see Section S1.





The ratios of INP concentrations based on Eq. 3 for freezing temperatures of -25, -20, and -15 °C, are shown in Figs. 4b, 4c, and 4d respectively. These ratios suggest that for the three samples when CCSEM-EDX measurements were performed, the $[INP(T)]_{MD}$ are higher than the $[INP(T)]_{SS}$ (ratios were between 10 to $10^3$, inclusive of errors, at -15, -20 and -25 °C), assuming the $n_s$-values used are applicable for the field studies reported here. These results also suggest that mineral
dust is a more important contributor to the INP population than sea spray aerosol for the times and locations corresponding to the CCSEM-EDX measurements.

### 3.3 Particle dispersion modelling

Figure 5a shows the averaged PES for the footprint layer for all samples combined and suggests that the source of INPs sampled during the campaign was local (i.e. the Canadian Arctic Archipelago, Baffin Bay, and eastern Greenland).
Figure 5b shows the averaged PES for the footprint layer for samples that had the highest INP concentrations (top 36 % of the samples) at -25 °C. Figure 5c shows the average PES for the footprint layer for samples that had the lowest INP concentrations (bottom 36 % of the samples) at -25 °C. A cut-off of 36 % was selected since no freezing was observed in 36 % of the samples at -25 °C. Figure 5d shows the ratio of the average PES for the highest INP concentrations to the average PES for all samples. Figure 5e shows the ratio of the average PES for the lowest INP concentrations to the average PES for
all samples. Previous work has shown that ratios of average PES are useful for identifying likely sources (Hirdman et al., 2010). Considering all figures together, the highest INP concentrations are associated with lower latitude regions such as the Hudson Bay area, eastern Greenland, or northwestern continental Canada. On the other hand, the lowest concentrations (below the detection limit at -25 °C) were often associated with regions further north and over Baffin Bay.

Figures 5f and 5g show maps of surface cover type (i.e. bare land, open water, sea ice, and snow cover) from the
first and last days of the campaign, respectively, based on data from the National Snow and Ice Data Center (NSIDC, 2008). The maps of surface coverage were combined with the PES values in the footprint layer from FLEXPART to determine the total residence time over each surface type for a given INP sample. Specifically, surface coverage data from the NSIDC was downloaded in GEOtiff format and converted to vector shapefiles. The fraction of each FLEXPART grid cell that was over each surface type category (e.g. bare land, open water) was then calculated using these vector shapefiles. The residence time
in a grid cell was then multiplied by the fraction of the cell in each surface type category, and then the results were summed over all grid cells to determine the relative time spent over each surface type.

Correlations between the total residence time over each surface type and the concentration of INPs for each sample at -15, -20, and -25 °C were then investigated (Table 1 and Fig S6). This correlation analysis showed statistically significant ($p < 0.05$) positive correlations between the total residence time over bare land in the footprint layer, and both $[INP(T)]$ at -
15 °C (R = 0.5) and -25 °C (R = 0.4). On the other hand, a statistically significant negative correlation was observed between the total residence time in the footprint layer over sea ice and both $[INP(T)]$ at -20 °C (R = -0.4) and -25 °C (R = -0.3). Furthermore, a statistically significant negative correlation was observed between the total residence time in the footprint layer over open water and $[INP(T)]$ at -20 °C (R = -0.4). These negative correlations can be explained by a stronger source



of INPs from bare land compared to sea ice or open water. Related, Bigg (1996) observed a correlation between INP concentrations measured in the high Arctic and the time since the sampled air mass was last in contact with open ocean (R = -0.54, p < 10⁻⁴). In contrast Bigg and Leck (2001) observed no correlation between INP concentrations and the time since the sampled air mass was last in contact with open ocean.

**4 Conclusions**

Concentrations of INPs in the marine boundary layer were measured at 28 different locations in the Canadian Arctic. Results showed that the concentrations of INPs are similar to concentrations measured in other marine environments.

For three collected samples the ratio of mineral dust surface area to sea spray surface area ranged from 0.03 to 0.09. Based on these ratios, and the ice active surface site densities of mineral dust and sea spray aerosol determined in previous
laboratory studies, mineral dust is a more important contributor to the INP population than sea spray aerosol for the samples analysed (ratios were between 10 to 10³, inclusive of errors, at -15 °C, -20 °C and -25 °C), assuming the ice active surface site densities of mineral dust and sea spray aerosol determined in laboratory studies are applicable to these field studies. This result suggests that INPs from mineral dust are more important contributors to the INP population than sea spray aerosol for the times and locations during sampling. Research has shown that some biological materials can act as efficient INPs (Ariya
et al., 2009; Conen et al., 2011; Creamean et al., 2013; Després et al., 2012; Haga et al., 2013, 2014; Hoose et al., 2010b; Iannone et al., 2011; Morris et al., 2013; Mortazavi et al., 2015; O'Sullivan et al., 2014; O′Sullivan et al., 2015; Pratt et al., 2009; Pummer et al., 2012; Spracklen and Heald, 2013; Stopelli et al., 2014; Tobo et al., 2013). More research is needed to determine the contribution of biological sources to the total INP population in the Arctic.

Particle dispersion modelling suggested that the INPs sampled in this study were likely not from long-range
transport. For the days where the [INP(T)] was high, a likely source was Hudson Bay area, eastern Greenland, or north-western continental Canada. For days where the [INP(T)] was low, the air mass spent more time over regions further north and over Baffin Bay.

Correlation analyses showed that there were statistically significant positive correlations between [INP(T)] at -15 and -25 °C and the time the air mass spent over land. Statistically significant negative correlations were observed between
[INP(T)] at -20 and -25 °C and the time the air mass spent over sea ice, and [INP(T)] at -20 °C and the time the air mass spent over open water. This correlation analysis together with the particle dispersion modelling provides further evidence that sea spray aerosol was likely not the major source of INPs during sampling.

As warming increases in the Arctic, more erodible soil will be exposed for longer periods of time (Huang et al., 2015). These results, together with our freezing results suggest that warming in the Arctic will increase concentrations of
INPs from mineral dust in the region, with possible implications for cloud properties and climate. Additional studies, including modelling and field studies are needed to quantify the importance of this feedback process for climate in the region.



**Acknowledgements**

We would like to thank the scientists, officers, and crew of the CCGS *Amundsen* for support during the expedition in 2014. We acknowledge the use of data products or imagery from the Land, Atmosphere Near real-time Capability for EOS (LANCE) system operated by the NASA/GSFC/Earth Science Data and Information System (ESDIS) with funding

5    provided by NASA/HQ.

The CCSEM-EDX analysis was performed at Environmental Molecular Sciences Laboratory at PNNL (Ringgold ID 47937 and 48781), a DOE Office of Science User Facility sponsored by the Office of Biological and Environmental Research. PNNL is operated for DOE by Battelle Memorial Institute under Contract DEAC06-76RL0 1830. We would also like to thank the Natural Sciences and Engineering Research Council of Canada and Fisheries and Oceans Canada for

10   funding.





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



| | -15 °C | | | -20 °C | | | -25 °C | | |
| --- | --- | --- | --- | --- | --- | --- | --- | --- | --- |
| | **R** | **p** | **n** | **R** | **p** | **n** | **R** | **p** | **n** |
| **Bare land residence time** | **0.5** | **0.008** | **27** | 0.3 | 0.058 | 27 | **0.4** | **0.033** | **27** |
| **Open water residence time** | -0.1 | 0.240 | 27 | **-0.4** | **0.023** | **27** | -0.3 | 0.054 | 27 |
| **Sea ice residence time** | -0.2 | 0.180 | 27 | **-0.4** | **0.028** | **27** | **-0.3** | **0.041** | **27** |
| **Snow cover residence time** | -0.2 | 0.162 | 27 | 0.2 | 0.183 | 27 | 0.0 | 0.449 | 27 |

**Table 1: Correlation coefficients (R), p and n values for correlation analysis between [INP] (L$^{-1}$) at -15, -20, and -25 °C and the time the air mass spent over different surface types within 0-300 m of the surface. Numbers in bold indicate correlations that are statistically significant ($p < 0.05$).**



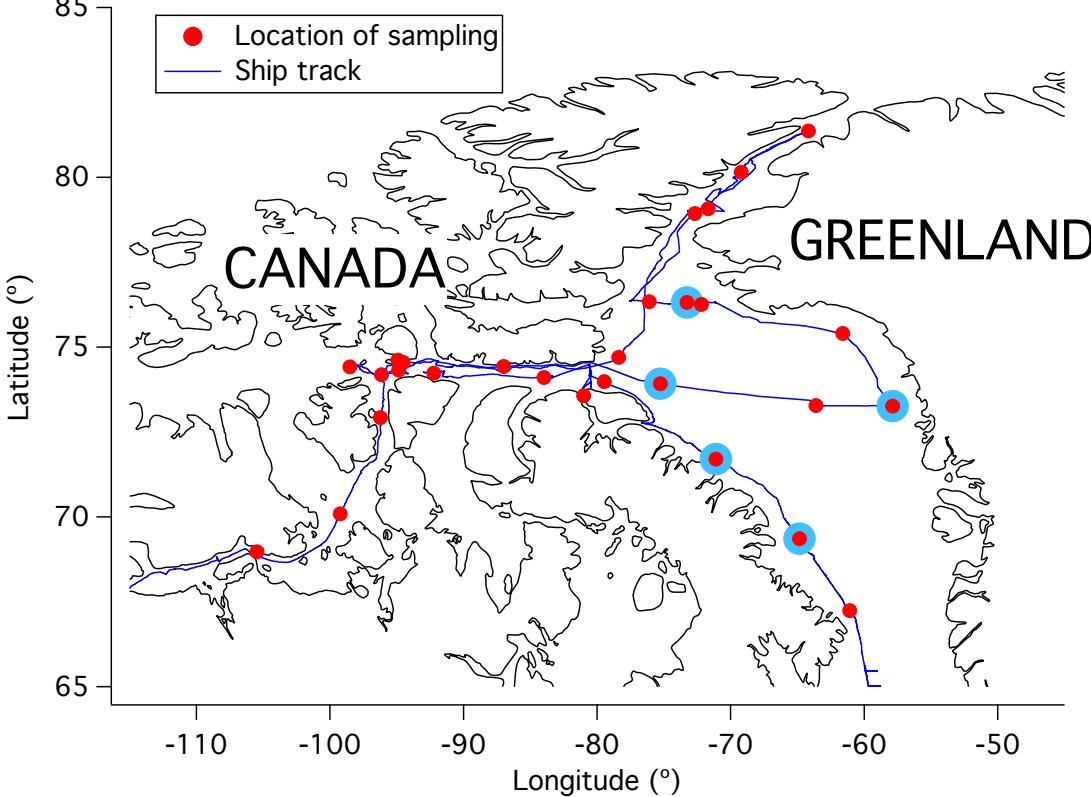

**Figure 1: Locations of sampling. Blue circles around the red dots indicate the locations of samples used in DeMott et al. (2016). Information on specific geographical coordinates is given in Supplemental Table S1.**



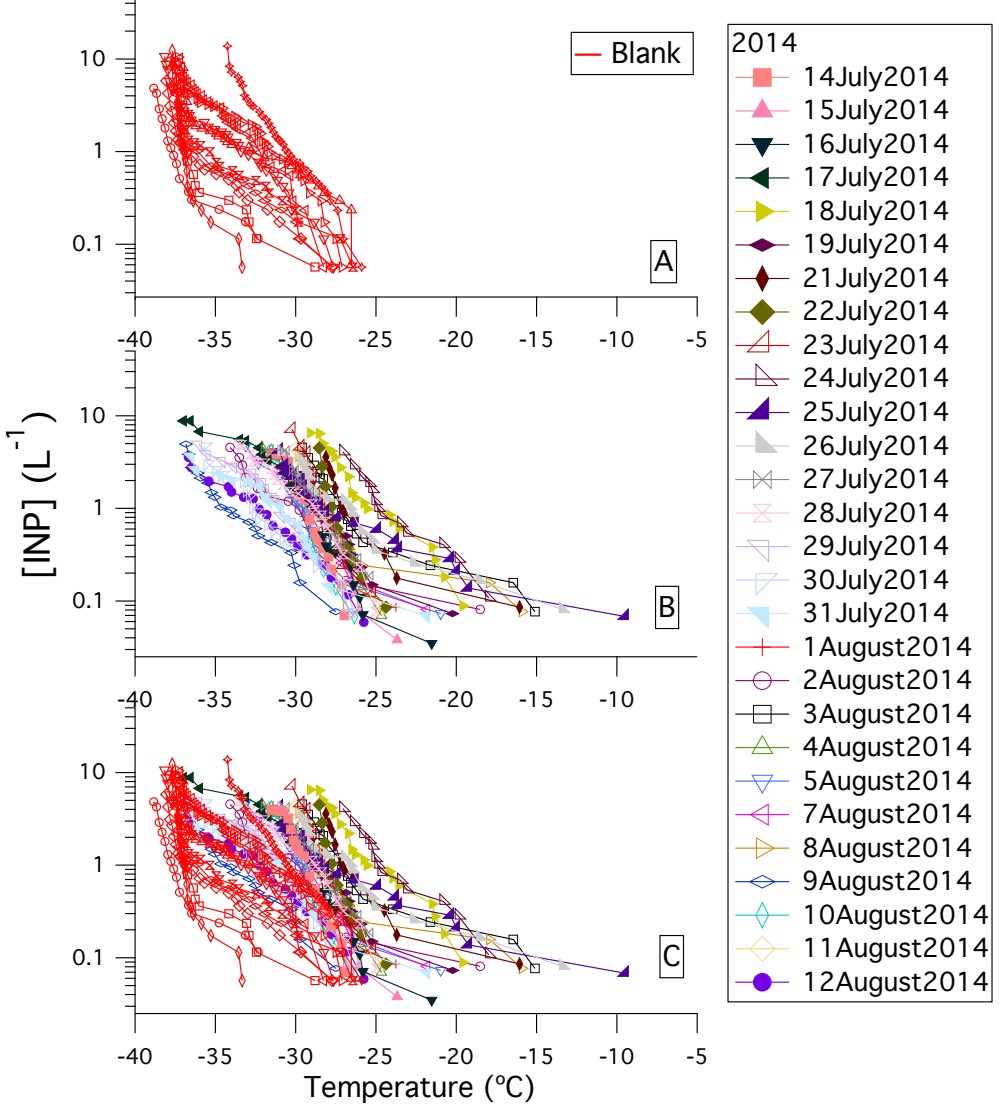

**Figure 2:** Plot of [INP] (L⁻¹) as a function of temperature (°C) for A) the blanks, B) the samples, and C) the blanks and samples. The [INP] (L⁻¹) of 11 blanks are shown in Panels A and C. Each blank was performed on a separate hydrophobic glass substrate. Error bars are not shown to improve the visibility of the data in the plot. Error bars in the x direction are ± 0.3 °C for each data point. Error bars in the y direction were calculated using nucleation statistics following Koop et al. (1997); the errors for our measured [INP] (L⁻¹) can be seen in Figure S5.



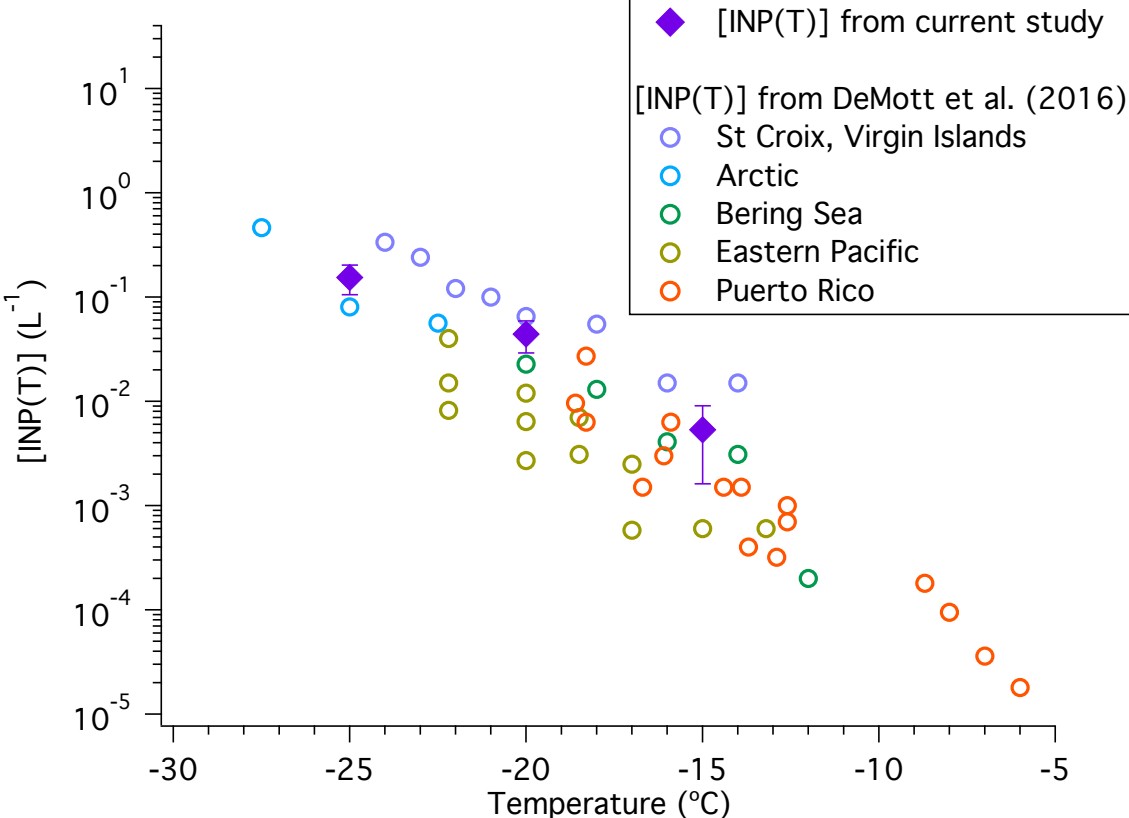

**Figure 3: Plot of [INP] (L⁻¹) as a function of temperature including a comparison to field results from several recent field studies in marine environments reported in DeMott et al. (2016). The purple diamonds represent averages of the data reported in this study at -15, -20, and -25 °C. Error bars are the standard error of the mean. Other coloured circles represent INP measurements from field studies in marine environments reported in DeMott et al. 2016, with the locations indicated in the legend.**





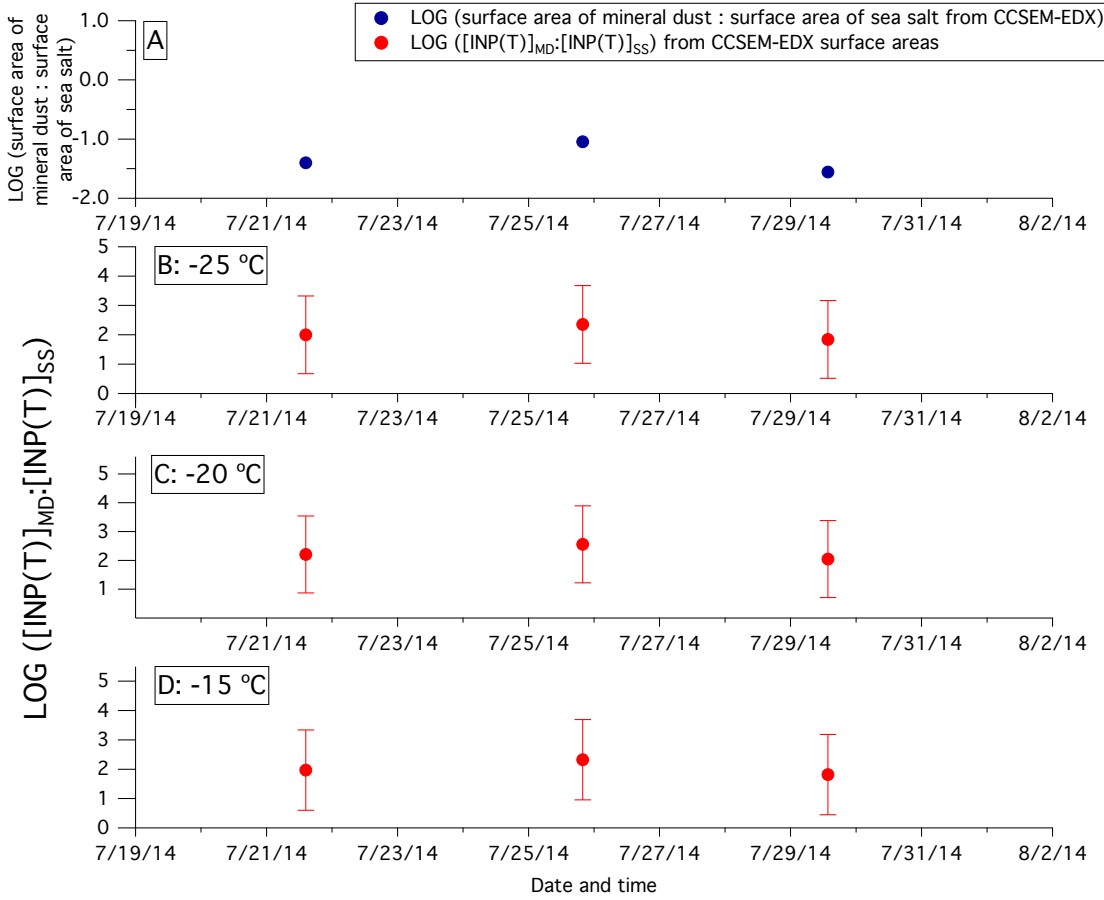

**Figure 4: A) Ratios of the surface area of mineral dust particles to the surface area of sea salt particles measured by CCSEM-EDX (blue circles). Ratios of predicted [INP(T)]$_{MD}$ to the predicted [INP(T)]$_{SS}$ calculated using CCSEM-EDX measurements (red circles) at B) -25 °C, C) -20 °C, and D) -15 °C.**



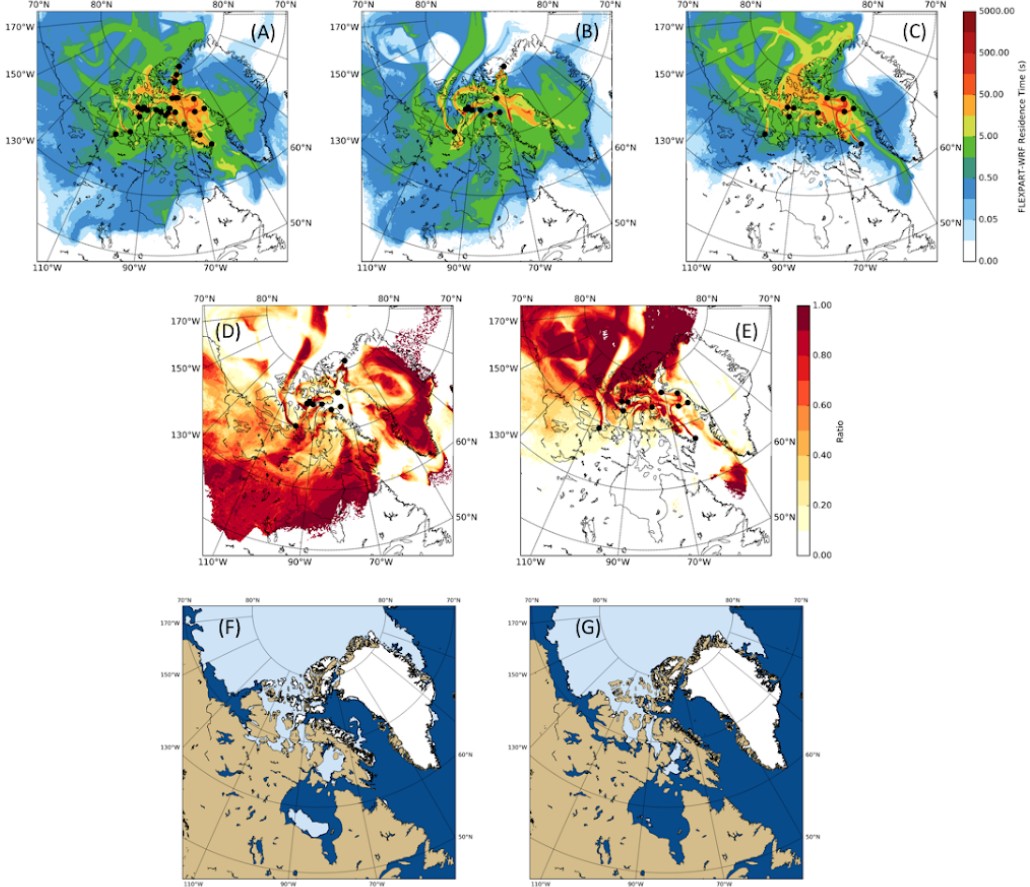

**Figure 5: Top row - Average FLEXPART-WRF footprint potential emission sensitivities (PES) plots for A) all sampling days, B) the 36 % of samples with the highest [INP(T)] (L⁻¹), and C) the 36 % of samples with the lowest [INP(T)] (L⁻¹). Black circles indicate the ship's position at the sampling mid-time. Middle row - Maps showing the ratios of D) plot B to plot A, and E) plot C to plot A. Bottom row - Maps showing the surface cover type on F) the first day of sampling (14ᵗʰ July), and G) the last day of sampling (12ᵗʰ August). Bare land, open water, sea ice, and snow cover are shown as beige, dark blue, light blue, and white, respectively. Note that for this study, lakes are included in the bare land category.**