# Peer review of "Ice nucleating particles in the marine boundary layer in the Canadian Arctic during summer 2014"

_Atmospheric Chemistry and Physics, 2018_

## Referee Comment (RC1) · Anonymous Referee #1 · 15 Sep 2018

**General Comments**

This paper represents a nice contribution to the literature, adding to the limited data on immersion freezing INP spectra in the Arctic region and emphasizing the dominant role that mineral dusts can play when overlain on pristine air masses that are otherwise representative of marine air. I was only curious about the use of a ratio calculation made on the basis of mineral and sea spray aerosol surface areas estimated by CC-SEM, rather than actually calculating a surface-active site density on the basis of the aerosol distribution to support what composition is most responsible for the INP activity. While the ratio approach is one to take, it would be good to see actual surface area

estimates in order to understand the consistency of the data with previous mineral dust parameterizations, rather than only assuming the validity a priori.

For example, only three samples were analyzed, understandable of course given the effort needed to perform the analyses of many particles for composition and size. Given this though, the unknown exact source of the dust, and the fact that what remains active after transport and any scavenging removal could differ from pure minerals tested in the laboratory, it could be interesting to know if the ice activity is truly consistent with dust parameterizations or is reduced and in a more competitive scenario with marine INPs. Otherwise, it is difficult to know if the inference of the dust dominance at all temperatures and loadings is as assumed. To know this, one might need to know mass fractions and surface areas very accurately. Hence, I suggest to add these actual values to the tables, and perhaps some discussion of alignment of data with the published or estimated parameterizations used to quantify assumed mineral influence. As for the estimates of marine INP contributions, the authors may know that an active site density parameterization is now in the literature (McCluskey et al., 2018), which seems to align quite well with the values they infer from published laboratory data.

I was a little curious about two factors in how the data were utilized. First, there is sparse data at -15 to -20 °C as emphasized temperatures. And I note on Page 9, lines 13-14, that even 36% of the samples had no INPs at -25 °C. Were "zero" or unresolvable data treated in some manner to come up with the average values plotted in Fig. 3? Also, a decision was made to not attempt to correct and discuss INPs for data at lower temperatures where background is present from the pure water. Was the correction simply too large in this region? Finally, I question whether time over marine areas would be expected to positively correlate with INP concentrations in any case. This is not obvious to me given the low strength of the marine source of INPs and the ready ability of terrestrial sources to dominate with any exposure to land emissions, which has been clear in some past studies.

Specific questions/comments for addressing before publication are listed below. The

paper is otherwise very well written, so these extra comments are limited to the topics overviewed above.

**Specific Comments**

1) Page 3, lines 17-20: Could the meaning of the wind directions noted be clarified? Do these imply from open water? Or assumed free from stack contamination? This actually touches on the topic of using remote ocean data and assuming marine only influence of course, about which not much is said.

2) Page 5, lines 18-19: Can you justify not considering the freezing of drops not on a spot as another sort of background freezing spectrum? Is it because you cannot be certain that some small particles did not migrate under this position on the slide?

3) Page 7, lines 3-5: This statement about the footprint layer confused me because I could not clearly distinguish how this differed from the earlier statement that the terminal point was 0 to 60 m above sea level. Please add, if possible.

4) Page 7, lines 19-20: One sample with "low" INP was selected, but I see only 4 or 5 samples in Fig. 2 that even have data at $-15\,°C$, so if three of these were used, then they are already not low INP samples I would judge. Is there a possible bias in the samples selected? After all, INP concentrations of 0.1 per liter at this temperature are fairly classical continental numbers in the first place. Your qualification about the conclusions being applicable only to the CCSEM cases on page 9 is duly noted. I wonder if you might comment about the influence of selection of samples for microscopy.

5) Page 7, lines 23-24: Please clarify if fully quantitative EDS analysis is performed to determine atomic percentages. That is the counts are interpreted quantitatively or qualitatively? Figure S4 shows actual concentration values as a means of interpreting mixtures as sea salt or dust, but there are no mixtures. It would be interesting to know how many of the dusts were salty, as processing may potentially alter their assumed behaviors (i.e., compared to parameterized dust).

6) Page 7, lines 27-28: Please also tabulate surface areas attributed to mineral dust and sea salt. It seems that one could also derive $n_s$ for each of these. This is critically important to the assumption that these can simply be applied in existing parameterizations that will then be turned into a ratio according to your Eq. 3.

7) Page 8, lines 18-21: Following from what I just said, if you have surface area, can't you compute $n_s$ and compare to parameterizations on the assumptions that particles are all SSA or all minerals, or use the mineral ratio to parse it out? Are the results consistent in any way with the parameterization or does the dust act differently than expected? Otherwise, by applying Niemand et al., you are making assumptions about the activity of the dust in these samples that may or may not be true. The same could be said about the marine spectrum, but the dust appears to dominate here, so is the most critical assumption.

8) Page 9: The correlation analyses are not especially impressive on first glance. Nevertheless, it seems to make sense that residence time over land would have a positive impact on INP concentrations, while time over water would not be expected to matter. This is the natural result when the land surface is a so much more powerful source by the amount that has been noted in past studies. The marine source would only be expected to show up when there is little or no land influence. Both sources would seem to depend more on wind conditions.

9) Page 10, lines 11-12: As noted, there is field evidence for marine INP ice active surface site densities in line with what is assumed for the exercise here.

10) Page 10, line 14: The comment about biological INPs is kind of a hanging thought. Are you inferring that the results are influenced by these? Your references suggest a range of source types including microbes or macromolecules it seems. While one might agree with the sentiment, it is not clear how it relates to the subject of this paper and why it only appears in the conclusions.

11) Page 10, lines 26-27: While the correlation analyses and dispersion modeling

support the role of mineral dust especially in case of higher loading (e.g., a few to 10% of surface area represented is a good amount of mineral dust), this does not indicate at what level marine INPs become important for lower INP concentrations does it? In a number of cases, INPs were apparently below detection limits, possibly consistent with typical surface areas and marine INP $n_s$, as well as with limited sample volumes assessed (limited warm temperature assessment). And again, any time spent over land would be expected to influence INPs strongly, while time spent over what might be a more constant and low INP source like the ocean would not be expected to correlate with INP concentrations. Those numbers may be relatively uniform independent of time spent over the marine source, but instead more correlated with marine conditions that influence emission rates. The fractional marine influence does not seem fully evaluated, as it would have required assessing more cases at the lower end of the spectrum of measurements I suspect.

**References**

McCluskey, C. S., Ovadnevaite, J., Rinaldi, M., Atkinson, J., Belosi, F., Ceburnis, D., et al. (2018). Marine and terrestrial organic ice-nucleating particles in pristine marine to continentally influenced Northeast Atlantic air masses. Journal of Geophysical Research: Atmospheres, 123. https://doi.org/10.1029/2017JD028033

---

## Referee Comment (RC2) · Anonymous Referee #2 · 24 Sep 2018

**Ice nucleating particles in Canadian Arctic sea-surface microlayer and bulk seawater.**

Victoria E. Irish, Sarah J. Hanna, Megan D. Willis, Swarup China, Jennie L. Thomas, Jeremy J. B. Wentzell, Ana Cirisan, Meng Si, W. Richard Leaitch, Jennifer G. Murphy, Jonathan P. D. Abbatt, Alexander Laskin, Eric Girard, Allan K. Bertram

https://doi.org/10.5194/acp-2018-735

**Summary**

This study researched the quantity, chemistry, and potential sources of immersion mode ice nucleating particles in the Canadian Arctic marine boundary layer during the summer of 2014. Aerosol samples were collected at 28 locations on a MOSSI impactor and then analyzed using a droplet freezing technique to quantify the concentration of INPs as a function of temperature. The ratio of mineral dust to sea spray particle surface area was quantified using EDX for three samples. These surface area ratios were then converted to active site density ratios, which revealed that mineral dust was the dominant INP type in the considered samples. Finally, the authors use the FLEXPART particle dispersion model to correlate INP concentration data with air mass back trajectories. This analysis suggests the source of INPs in the artic marine boundary layer are relatively local mineral or soil dusts.

**General Comments**

This study makes a substantial contribution to the field by presenting an impressive amount of immersion mode INP concentration data for the Arctic. The locality of the field work is very timely, and the additional analyses on particle composition and back trajectories nicely support the authors' main goal of quantifying immersion mode INP concentrations. The paper is well suited for publication in ACP. Below, I provide a few questions and comments to strengthen the paper and clarify its results.

**Note on INP Nomenclature:**

Please try to be clearer throughout the text that you considered the immersion freezing mode of ice nucleation. This is especially important given your results that mineral dust was an important source of particulate matter in your samples: mineral dust can activate in the deposition freezing mode at the temperatures considered here (e.g. Hoose and Mohler, 2012), but the mode of freezing investigated in this study isn't even included in the abstract.

**Specific Comments**

1. **Page 2, Line 9:** Please clarify here that sea spray particles actually vary widely in composition, depending on mechanism of formation. For instance, film rupture particles are organic enriched compared to jet droplet particles (e.g. Wang et al. 2017). This will highlight the importance of considering both microlayer and bulk seawater samples, as you suggest in the following sentence.
2. **Page 2, Line 11:** For completeness, the authors may also consider citing Huang et al. 2018 here. This paper follows up on Burrows et al. 2013 by discussing the sensitivity of the relative contribution of sea spray to INP concentrations at high latitudes to a model's choice of active site density parameterizations.

3. **Page 3, Line 15:** Were the metrological data collected with on-ship sensors? If so, can some estimate of the height of these sensors above sea level be given?

4. **Page 4, Line 8:** Has the transmission efficiency of the louvered total suspended particulate (TSP) inlet been quantified before? For example, it sounds similar to an inlet characterized by Kenny et al. 2005. If the exact loss percentages have not been measured, that's okay; but the authors should note that larger particles (like mineral dust) may be more prone to impaction and thus be undersampled on the MOSSI slides.

5. **Page 6, Line 7:** Was the height of the chemical ionization mass spectrometer inlet at a similar height to the inlet of the MOSSI?

6. **Page 7, Line 17:** The authors state that only particles at the edge of the spots were analyzed with EDX, but how are they sure these particles are representative of the bulk of particles directly below the micro-orifices? For example, the authors mention the particle rebound effect (Page 4, Line 12). Can the authors discuss if perhaps mineral dust is more prone to rebound than the deliquesced salty particles, and therefore may be more likely to end up at the edge of the spot where EDX was performed?

7. **Page 7, Line 22:** "First, the atomic percentages of each particle were determined from EDX spectra." Can the authors clarify if only one discrete spot of each particle was examined, or if the EDX data represents an average signal from a raster scan?

8. **Page 7, Line 23:** Do the authors have any way to estimate what fraction of particles were internally mixed; i.e. mineral dust coated with sea salt?

9. **Page 8, Line 4:** How were blanks prepared and treated? Were they fresh slides out of the package? Or were they treated the same way as the sampled slides, i.e. placed within the MOSSI only without turning on the pump? The latter would better account for contamination in the sample handling and preparation process; but in either case, more information is needed here.

10. **Page 8, Line 17:** "The two samples corresponding to high [INP(T)] were collected on July 21st and 25th." As I read Figure 2, these are among the highest but not the highest INP concentration days. Can the authors provide any more reasoning behind why these particular days were chosen for compositional analysis? How typical was the meteorology on these days? Or perhaps they were chosen to be evenly spaced in time and location? Or was the choice random?

11. **Page 8, Line 14:** The jump to biological particles is a bit of a non-sequitur. You might elaborate on how biological activity can influence sea spray particle composition, which in turn effects INP activity.

12. **Page 8, Conclusions:** Can the authors discuss their findings that mineral dust immersion INPs dominated over sea spray INPs in the context of Irish et al. 2017, which found immersion mode INPs to be abundant in the seawater from this region? E.g. perhaps the INPs were present in the seawater but were never aerosolized because it wasn't windy enough to generated sea spray. This also ties in with Reviewer 1's comment that time spent over open ocean may not correlate with sea spray INP concentration if sea spray particles are not being formed. Then, include a discussion on whether these conditions are typical: how do your wind speeds compare with average (intra- and interannual) wind speeds in the region? Briefly delving into a reanalysis dataset (e.g. ECMWF ERA-Interim) might help you to explore this question in depth, or you could look for historical observational data. This will require a little bit more work on your end, but it would strengthen your analysis immensely by allowing you to hypothesize whether your findings are typical.

13. **Page 20, Table 1:** Please include in the table caption the length of the back-trajectory (7 days).
14. **Page 21, Figure 1:** Can you also denote in this figure where the samples for chemical/EDX analysis were collected? That will help us visualize the potential sources of these particles in light of the data in Figure 5. Otherwise, this figure looks great!
15. **Page 23, Figure 3:** Please also show the min/max INP concentration at each temperature (in addition to the average and standard deviation you've already shown) to provide more information on the degree of variability. Are the data from DeMott et al. 2015 averages, or discrete measurements?
16. **Page 25, Figure 5:** Please increase the font size in this figure to something legible.
17. **Supplement, Page 9, Figure S6:** Please include units ($L^{-1}$) for INP concentrations either in the axis label or the figure caption.

**References**:

Hoose, C. and Möhler, O., 2012. Heterogeneous ice nucleation on atmospheric aerosols: a review of results from laboratory experiments, Atmospheric Chemistry and Physics, 12, 9817–9854.

Huang, W.T.K., Ickes, L., Tegen, I., Rinaldi, M., Ceburnis, D. and Lohmann, U., 2018. Global relevance of marine organic aerosol as ice nucleating particles. *Atmospheric Chemistry & Physics*, *18*(15).

Kenny, L., Beaumont, G., Gudmundsson, A., Thorpe, A. and Koch, W., 2005. Aspiration and sampling efficiencies of the TSP and louvered particulate matter inlets. *Journal of Environmental Monitoring*, *7*(5), pp.481-487.

Wang, X., Deane, G.B., Moore, K.A., Ryder, O.S., Stokes, M.D., Beall, C.M., Collins, D.B., Santander, M.V., Burrows, S.M., Sultana, C.M. and Prather, K.A., 2017. The role of jet and film drops in controlling the mixing state of submicron sea spray aerosol particles. *Proceedings of the National Academy of Sciences*, *114*(27), pp.6978-6983.

---

## Author Comment (AC1) · 13 Dec 2018

Prof. Daniel Cziczo
Co-Editor of Atmospheric Chemistry and Physics

Dear Dan,

Listed below are our responses to the comments from the reviewers of our manuscript. For clarity and visual distinction, the referee comments or questions are listed here in black and are preceded by bracketed, italicized numbers (e.g. *[1]*). Authors' responses are in red below each referee statement with matching numbers (e.g. *[A1]*).  We thank the reviewers for carefully reading our manuscript and for their helpful suggestions!

Sincerely,

Allan Bertram
Professor of Chemistry
University of British Columbia

**Anonymous Referee #1**

**General Comments**

This paper represents a nice contribution to the literature, adding to the limited data on immersion freezing INP spectra in the Arctic region and emphasizing the dominant role that mineral dusts can play when overlain on pristine air masses that are otherwise representative of marine air. I was only curious about the use of a ratio calculation made on the basis of mineral and sea spray aerosol surface areas estimated by CCSEM, rather than actually calculating a surface-active site density on the basis of the aerosol distribution to support what composition is most responsible for the INP activity. While the ratio approach is one to take, it would be good to see actual surface area estimates in order to understand the consistency of the data with previous mineral dust parameterizations, rather than only assuming the validity a priori. For example, only three samples were analyzed, understandable of course given the effort needed to perform the analyses of many particles for composition and size. Given this though, the unknown exact source of the dust, and the fact that what remains active after transport and any scavenging removal could differ from pure minerals tested in the laboratory, it could be interesting to know if the ice activity is truly consistent with dust parameterizations or is reduced and in a more competitive scenario with marine INPs. Otherwise, it is difficult to know if the inference of the dust dominance at all temperatures and loadings is as assumed. To know this, one might need to know mass fractions and surface areas very accurately. Hence, I suggest to add these actual values to the tables, and perhaps some discussion of alignment of data with the published or estimated parameterizations used to quantify assumed mineral influence.

As for the estimates of marine INP contributions, the authors may know that an active site density parameterization is now in the literature (McCluskey et al., 2018), which seems to align quite well with the values they infer from published laboratory data. I was a little curious about two factors in how the data were utilized. First, there is sparse data at -15 to -20 °C as emphasized temperatures. And I note on Page 9, lines 13-14, that even 36% of the samples had no INPs at -25 °C. Were "zero" or unresolvable data treated in some manner to come up with the average values plotted in Fig. 3? Also, a decision was made to not attempt to correct and discuss INPs for data at lower temperatures where background is present from the pure water. Was the correction simply too large in this region? Finally, I question whether time over marine areas would be expected to positively correlate with INP concentrations in any case. This is not obvious to me given the low strength of the marine source of INPs and the ready ability of terrestrial sources to dominate with any exposure to land emissions, which has been clear in some past studies. Specific questions/comments for addressing before publication are listed below. The paper is otherwise very well written, so these extra comments are limited to the topics overviewed above.

**Specific Comments**

*[1] Page 3, lines 17-20: Could the meaning of the wind directions noted be clarified? Do these imply from open water? Or assumed free from stack contamination? This actually touches on the topic of using remote ocean data and assuming marine only influence of course, about which not much is said.*

*[A1] The wind directions noted correspond to conditions assumed to be free from stack contamination. To address the referee's comment this information will be added to the revised manuscript. Specifically the following text will be added to the manuscript (Section 2.1):*

*"The data reported in DeMott et al. (2016) only included sites in Baffin Bay, days where it did not rain and conditions when the samples were assumed to have reduced exposure to ship smokestack contamination i.e. when the apparent wind direction measured on the ship was between 0-90 degrees or 270-360 degrees, where 0/360 corresponds to the bow of ship"*

*[2] Page 5, lines 18-19: Can you justify not considering the freezing of drops not on a spot as another sort of background freezing spectrum? Is it because you cannot be certain that some small particles did not migrate under this position on the slide?*

*[A2] Correct - we cannot be certain that some small particles do not migrate under this position on the slide. As stated on Page 5, Lines 19-21, "We assumed these relatively rare occurrences were due to particles < 0.18 µm in diameter that were not focused into spots or due to rebound of a small fraction of the particles off the hydrophobic glass slides". We do not consider this freezing as another sort of background freezing spectrum since no freezing from the blanks was observed at temperatures ≥-25 °C.*

*[3]* Page 7, lines 3-5: This statement about the footprint layer confused me because I could not clearly distinguish how this differed from the earlier statement that the terminal point was 0 to 60 m above sea level. Please add, if possible.

*[A3] At the start of the simulation the particles are released in a box and are followed backwards in time. For our study the box had the dimensions 100 x 100 m in the horizontal and from 0 to 60 m in the vertical. This initial box is only relevant for the release, after which the particles are free to move in all dimensions. After release of the particles, they are followed backwards in time to determine potential emission sensitivity (PES) plots. Since we are concerned with emissions from the surface we determined PES plots for the footprint layer (0 - 300 m).*

*The authors will add text to the manuscript to clarify that 0 - 60 m in the vertical only relates to initial simulation conditions.*

*[4]* Page 7, lines 19-20: One sample with "low" INP was selected, but I see only 4 or 5 samples in Fig. 2 that even have data at -15 ∘C, so if three of these were used, then they are already not low INP samples I would judge. Is there a possible bias in the samples selected? After all, INP concentrations of 0.1 per liter at this temperature are fairly classical continental numbers in the first place. Your qualification about the conclusions being applicable only to the CCSEM cases on page 9 is duly noted. I wonder if you might comment about the influence of selection of samples for microscopy.

*[A4] We chose to use two samples that showed high INP concentrations and one sample with a low INP concentration. The 29$^{th}$ July sample did not display freezing activity above -25 °C, which is why it was considered a low INP concentration day. We explored two days with higher INP concentrations, the 25$^{th}$ July and the 21$^{st}$ July. These two days were random choices. We will adjust the text in section 3.2 to read:*

*"The two samples corresponding to high [INP(T)] were collected on July 21$^{st}$ and 25$^{th}$, these days were chosen randomly out of the samples that showed freezing activity at -20 °C and -25 °C. The sample corresponding to a low [INP(T)] was collected on July 29$^{th}$, this day was chosen randomly out of the samples that did not display any freezing activity above -25 °C."*

*[5]* Page 7, lines 23-24: Please clarify if fully quantitative EDS analysis is performed to determine atomic percentages. That is the counts are interpreted quantitatively or qualitatively? Figure S4 shows actual concentration values as a means of interpreting mixtures as sea salt or dust, but there are no mixtures. It would be interesting to know how many of the dusts were salty, as processing may potentially alter their assumed behaviors (i.e., compared to parameterized dust).

*[A5] A fully quantitative EDX analysis was performed to determine atomic percentages. For the current analysis, we combined particles classified as mineral dust and mixtures of mineral dust and sea salt together, as illustrated in Figure S4. If we used the*

*classification scheme by Derimian et al. (2017) that explicitly classifies particles as sea salt, mineral dust, and mixed sea salt/mineral dust, for the three samples analysed, no particles are classified as mixed sea salt/mineral dust. This will be stated in the manuscript.*

*[6]* Page 7, lines 27-28: Please also tabulate surface areas attributed to mineral dust and sea salt. It seems that one could also derive ns for each of these. This is critically important to the assumption that these can simply be applied in existing parameterizations that will then be turned into a ratio according to your Eq. 3.

*[A6] To address the referee's comment the surface areas attributed to mineral dust and sea salt will be added in Table S2 in the Supplemental. Derivation of $n_s$ is discussed below.*

*[7]* Page 8, lines 18-21: Following from what I just said, if you have surface area, can't you compute ns and compare to parameterizations on the assumptions that particles are all SSA or all minerals, or use the mineral ratio to parse it out? Are the results consistent in any way with the parameterization or does the dust act differently than expected? Otherwise, by applying Niemand et al., you are making assumptions about the activity of the dust in these samples that may or may not be true. The same could be said about the marine spectrum, but the dust appears to dominate here, so is the most critical assumption.

*[A7] To address the referee's comment we will add calculations of $n_s$-values of mineral dust to the revised manuscript and compare the calculated $n_s$-values to data in the literature for mineral dust.*

*[8]* Page 9: The correlation analyses are not especially impressive on first glance. Nevertheless, it seems to make sense that residence time over land would have a positive impact on INP concentrations, while time over water would not be expected to matter. This is the natural result when the land surface is a so much more powerful source by the amount that has been noted in past studies. The marine source would only be expected to show up when there is little or no land influence. Both sources would seem to depend more on wind conditions.

*[A8] The authors agree with the referee.*

*[9]* Page 10, lines 11-12: As noted, there is field evidence for marine INP ice active surface site densities in line with what is assumed for the exercise here.

*[A9] In the revised manuscript we will reference the field evidence for marine INP ice active surface site densities.*

*[10]* Page 10, line 14: The comment about biological INPs is kind of a hanging thought. Are you inferring that the results are influenced by these? Your references suggest a range of source types including microbes or macromolecules it seems. While one might

agree with the sentiment, it is not clear how it relates to the subject of this paper and why it only appears in the conclusions.

*[A10] We will remove the comment about biological INPs.*

*[11]* Page 10, lines 26-27: While the correlation analyses and dispersion modeling support the role of mineral dust especially in case of higher loading (e.g., a few to 10% of surface area represented is a good amount of mineral dust), this does not indicate at what level marine INPs become important for lower INP concentrations does it? In a number of cases, INPs were apparently below detection limits, possibly consistent with typical surface areas and marine INP ns, as well as with limited sample volumes assessed (limited warm temperature assessment). And again, any time spent over land would be expected to influence INPs strongly, while time spent over what might be a more constant and low INP source like the ocean would not be expected to correlate with INP concentrations. Those numbers may be relatively uniform independent of time spent over the marine source, but instead more correlated with marine conditions that influence emission rates. The fractional marine influence does not seem fully evaluated, as it would have required assessing more cases at the lower end of the spectrum of measurements I suspect.

*[A11] The authors agree with the referee.*

*We will modify this discussion from the following:* "This correlation analysis together with the particle dispersion modelling provides further evidence, that sea spray aerosol was likely not the major source of INPs during sampling."

To:

"This correlation analysis together with the particle dispersion modelling suggests that sea spray aerosol was likely not the major source of INPs during sampling, at least not when INP concentrations were high. Sea spray aerosol may still have played a role when the INP concentrations were low during sampling."

**References**

McCluskey, C. S., Ovadnevaite, J., Rinaldi, M., Atkinson, J., Belosi, F., Ceburnis, D., et al. (2018). Marine and terrestrial organic ice-nucleating particles in pristine marine to continentally influenced Northeast Atlantic air masses. Journal of Geophysical Research: Atmospheres, 123. https://doi.org/10.1029/2017JD028033

**Anonymous Referee #2**

**Summary**

This study researched the quantity, chemistry, and potential sources of immersion mode ice nucleating particles in the Canadian Arctic marine boundary layer during the summer of 2014. Aerosol samples were collected at 28 locations on a MOSSI impactor and then analyzed using a droplet freezing technique to quantify the concentration of INPs as a function of temperature. The ratio of mineral dust to sea spray particle surface area was quantified using EDX for three samples. These surface area ratios were then converted to active site density ratios, which revealed that mineral dust was the dominant INP type in the considered samples. Finally, the authors use the FLEXPART particle dispersion model to correlate INP concentration data with air mass back trajectories. This analysis suggests the source of INPs in the arctic marine boundary layer are relatively local mineral or soil dusts.

**General Comments**

This study makes a substantial contribution to the field by presenting an impressive amount of immersion mode INP concentration data for the Arctic. The locality of the field work is very timely, and the additional analyses on particle composition and back trajectories nicely support the authors' main goal of quantifying immersion mode INP concentrations. The paper is well suited for publication in ACP. Below, I provide a few questions and comments to strengthen the paper and clarify its results.

**Note on INP Nomenclature**

[12] Please try to be clearer throughout the text that you considered the immersion freezing mode of ice nucleation. This is especially important given your results that mineral dust was an important source of particulate matter in your samples: mineral dust can activate in the deposition freezing mode at the temperatures considered here (e.g. Hoose and Mohler, 2012), but the mode of freezing investigated in this study isn't even included in the abstract.

[A12] We will clarify the mode of ice nucleation to be immersion freezing throughout the manuscript.

**Specific Comments**

[13] Page 2, Line 9: Please clarify here that sea spray particles actually vary widely in composition, depending on mechanism of formation. For instance, film rupture particles are organic enriched compared to jet droplet particles (e.g. Wang et al. 2017). This will highlight the importance of considering both microlayer and bulk seawater samples, as you suggest in the following sentence.

[A13] We will clarify that sea spray particles vary widely in composition depending on mechanism of formation. This will be done by amending the text to:

"Sea spray aerosol is generated by a bubble bursting mechanism at the ocean surface (Blanchard, 1964) and varies widely in composition, depending on the mechanism of

formation. For example, particles from film rupture are enriched in organics compared to particles from jet droplets (Wang et al., 2017)."

[14] Page 2, Line 11: For completeness, the authors may also consider citing Huang et al. 2018 here. This paper follows up on Burrows et al. 2013 by discussing the sensitivity of the relative contribution of sea spray to INP concentrations at high latitudes to a model's choice of active site density parameterizations.

[A14] We will add the citation Huang et al. (2018) to the manuscript.

[15] Page 3, Line 15: Were the metrological data collected with on-ship sensors? If so, can some estimate of the height of these sensors above sea level be given?

[A15] The meteorological data was collected with on-ship sensors. We will add the height of the sensors used to measure RH, wind speed, and temperature.

[16] Page 4, Line 8: Has the transmission efficiency of the louvered total suspended particulate (TSP) inlet been quantified before? For example, it sounds similar to an inlet characterized by Kenny et al. 2005. If the exact loss percentages have not been measured, that's okay; but the authors should note that larger particles (like mineral dust) may be more prone to impaction and thus be undersampled on the MOSSI slides.

[A16] The transmission efficiency of the louvered total suspended particulate inlet has been quantified previously (Kenny et al., 2005). Based on this previous study, the transmission efficiency of 10 $\mu$m particles through the inlet is $\geq$ 90 %. To address the referee's comments, this information will be added to the revised manuscript.

[17] Page 6, Line 7: Was the height of the chemical ionization mass spectrometer inlet at a similar height to the inlet of the MOSSI?

[A17] The authors will add the height of the CIMS to the manuscript.

[18] Page 7, Line 17: The authors state that only particles at the edge of the spots were analyzed with EDX, but how are they sure these particles are representative of the bulk of particles directly below the micro-orifices? For example, the authors mention the particle rebound effect (Page 4, Line 12). Can the authors discuss if perhaps mineral dust is more prone to rebound than the deliquesced salty particles, and therefore may be more likely to end up at the edge of the spot where EDX was performed?

[A18] The particles on the edge of the spots were directly under the nozzles, meaning they were part of the spot. We assume the composition of the edge of the spot is the same as the whole spot; however, we are unable to confirm this assumption. In the revised manuscript we will clearly state this assumption by adding the following:

"We assume the composition of the edge of the spot is the same as the composition of the whole spot, although we are unable to confirm this assumption."

*[19]* Page 7, Line 22: "First, the atomic percentages of each particle were determined from EDX spectra." Can the authors clarify if only one discrete spot of each particle was examined, or if the EDX data represents an average signal from a raster scan?

*[A19] To address the referee's comment the authors will clarify this point in the manuscript by adding the following sentence:*

"The EDX data for an individual particle represents an average signal from a raster scan."

*[20]* Page 7, Line 23: Do the authors have any way to estimate what fraction of particles were internally mixed; i.e. mineral dust coated with sea salt?

*[A20] See [A5] above.*

*[21]* Page 8, Line 4: How were blanks prepared and treated? Were they fresh slides out of the package? Or were they treated the same way as the sampled slides, i.e. placed within the MOSSI only without turning on the pump? The latter would better account for contamination in the sample handling and preparation process; but in either case, more information is needed here.

*[A21] The authors will include more information on how the blanks were prepared and treated in the manuscript.*

*[22]* Page 8, Line 17: "The two samples corresponding to high [INP(T)] were collected on July 21st and 25th." As I read Figure 2, these are among the highest but not the highest INP concentration days. Can the authors provide any more reasoning behind why these particular days were chosen for compositional analysis? How typical was the meteorology on these days? Or perhaps they were chosen to be evenly spaced in time and location? Or was the choice random?

*[A22] The authors analysed two days with higher INP concentrations, the 25th July and the 21st July. These two days were random choices. The authors have adjusted the text in section 3.2 to read:*

"The two samples corresponding to high [INP(T)] were collected on July 21st and 25th, these days were chosen randomly out of the samples that showed freezing activity at -20 °C and -25 °C . The sample corresponding to a low [INP(T)] was collected on July 29th, this day was chosen randomly out of the samples that did not display any freezing activity above -25 °C."

*The 21st and 25th had 5 m/s and 3 m/s average wind speeds, respectively. The average wind speed during sampling was 5 m/s. The samples were not chosen because of the meteorology.*

*[23]* Page 10, Line 14: The jump to biological particles is a bit of a non-sequitur. You might elaborate on how biological activity can influence sea spray particle composition, which in turn effects INP activity.

*[A23] We will take this statement out of the manuscript.*

*[24]* Page 10, Conclusions: Can the authors discuss their findings that mineral dust immersion INPs dominated over sea spray INPs in the context of Irish et al. 2017, which found immersion mode INPs to be abundant in the seawater from this region? E.g. perhaps the INPs were present in the seawater but were never aerosolized because it wasn't windy enough to generated sea spray. This also ties in with Reviewer 1's comment that time spent over open ocean may not correlate with sea spray INP concentration if sea spray particles are not being formed. Then, include a discussion on whether these conditions are typical: how do your wind speeds compare with average (intra- and interannual) wind speeds in the region? Briefly delving into a reanalysis dataset (e.g. ECMWF ERA-Interim) might help you to explore this question in depth, or you could look for historical observational data. This will require a little bit more work on your end, but it would strengthen your analysis immensely by allowing you to hypothesize whether your findings are typical.

*[A24] The authors will address the referees comment by adding the following discussion in the manuscript:*

*"Previous studies have shown that INPs are ubiquitous in Arctic seawater (Irish et al., 2017; Wilson et al., 2015). Our results show that INPs in the Arctic seawater are not emitted at a high enough rate to compete with mineral dust, at least for the samples with high INP concentrations. The flux of sea spray aerosol to the atmosphere is a function of the wind speed. The average minute wind speed during sampling (5.5 m s$^{-1}$ with 10$^{th}$ and 90$^{th}$ percentiles of 2.1 and 10.0 m s$^{-1}$) was similar to the average minute wind speed during the whole campaign (5.2 m s$^{-1}$ with 10$^{th}$ and 90$^{th}$ percentiles of 1.5 and 9.3 m s$^{-1}$). In addition, our average wind speeds were similar to historical monthly average wind speed data from Alert, NU, Canada (3.6 m s$^{-1}$ in July and 3.3 m s$^{-1}$ in August; climate.weather.gc.ca; climate ID: 2400300). The influence of sea spray aerosol may be more important during periods of higher wind speeds. On the other hand, high wind speeds are also likely to increase the flux of mineral dust from local sources."*

*[25]* Page 20, Table 1: Please include in the table caption the length of the back-trajectory (7 days).

*[A25] The length of the back-trajectory will be included in the caption of Table 1.*

*[26]* Page 21, Figure 1: Can you also denote in this figure where the samples for chemical/EDX analysis were collected? That will help us visualize the potential sources of these particles in light of the data in Figure 5. Otherwise, this figure looks great!

*[A26] The authors will indicate which sampling locations correspond to samples used for CCSEM-EDX analysis.*

*[27]* Page 23, Figure 3: Please also show the min/max INP concentration at each temperature (in addition to the average and standard deviation you've already shown) to provide more information on the degree of variability. Are the data from DeMott et al. 2015 averages, or discrete measurements?

*[A27] The author will clarify that the data from DeMott et al. (2016) are discrete measurements, and the min/max INP concentration at each temperature will also be added to the figure.*

*[28]* Page 25, Figure 5: Please increase the font size in this figure to something legible.

*[A28] The font size in Figure 5 will be increased.*

*[29]* Supplement, Page 9, Figure S6: Please include units ($L^{-1}$) for INP concentrations either in the axis label or the figure caption.

*[A29] ($L^{-1}$) will be added to the axis label in Figure S6 (now Figure S7).*

**References**

Hoose, C. and Möhler, O., 2012. Heterogeneous ice nucleation on atmospheric aerosols: a review of results from laboratory experiments, Atmospheric Chemistry and Physics, 12, 9817– 9854.

Huang, W.T.K., Ickes, L., Tegen, I., Rinaldi, M., Ceburnis, D. and Lohmann, U., 2018. Global relevance of marine organic aerosol as ice nucleating particles. Atmospheric Chemistry & Physics, 18(15).

Kenny, L., Beaumont, G., Gudmundsson, A., Thorpe, A. and Koch, W., 2005. Aspiration and sampling efficiencies of the TSP and louvered particulate matter inlets. Journal of Environmental Monitoring, 7(5), pp.481-487.

Wang, X., Deane, G.B., Moore, K.A., Ryder, O.S., Stokes, M.D., Beall, C.M., Collins, D.B., Santander, M.V., Burrows, S.M., Sultana, C.M. and Prather, K.A., 2017. The role of jet and film drops in controlling the mixing state of submicron sea spray aerosol particles. Proceedings of the National Academy of Sciences, 114(27), pp.6978-6983.

---

## Author Response (AR1)

Prof. Daniel Cziczo
Co-Editor of Atmospheric Chemistry and Physics

Dear Dan,

Listed below are our responses to the comments from the reviewers of our manuscript. For clarity and visual distinction, the referee comments or questions are listed here in black and are preceded by bracketed, italicized numbers (e.g. *[1]*). Authors' responses are in red below each referee statement with matching numbers (e.g. *[A1]*). We thank the reviewers for carefully reading our manuscript and for their helpful suggestions!

Sincerely,

Allan Bertram
Professor of Chemistry
University of British Columbia

**Anonymous Referee #1**

**General Comments**

This paper represents a nice contribution to the literature, adding to the limited data on immersion freezing INP spectra in the Arctic region and emphasizing the dominant role that mineral dusts can play when overlain on pristine air masses that are otherwise representative of marine air. I was only curious about the use of a ratio calculation made on the basis of mineral and sea spray aerosol surface areas estimated by CCSEM, rather than actually calculating a surface-active site density on the basis of the aerosol distribution to support what composition is most responsible for the INP activity. While the ratio approach is one to take, it would be good to see actual surface area estimates in order to understand the consistency of the data with previous mineral dust parameterizations, rather than only assuming the validity a priori. For example, only three samples were analyzed, understandable of course given the effort needed to perform the analyses of many particles for composition and size. Given this though, the unknown exact source of the dust, and the fact that what remains active after transport and any scavenging removal could differ from pure minerals tested in the laboratory, it could be interesting to know if the ice activity is truly consistent with dust parameterizations or is reduced and in a more competitive scenario with marine INPs. Otherwise, it is difficult to know if the inference of the dust dominance at all temperatures and loadings is as assumed. To know this, one might need to know mass fractions and surface areas very accurately. Hence, I suggest to add these actual values to the tables, and perhaps some discussion of alignment of data with the published or estimated parameterizations used to quantify assumed mineral influence.

As for the estimates of marine INP contributions, the authors may know that an active site density parameterization is now in the literature (McCluskey et al., 2018), which seems to align quite well with the values they infer from published laboratory data. I was a little curious about two factors in how the data were utilized. First, there is sparse data at -15 to -20 ∘C as emphasized temperatures. And I note on Page 9, lines 13-14, that even 36% of the samples had no INPs at -25 ∘C. Were "zero" or unresolvable data treated in some manner to come up with the average values plotted in Fig. 3? Also, a decision was made to not attempt to correct and discuss INPs for data at lower temperatures where background is present from the pure water. Was the correction simply too large in this region? Finally, I question whether time over marine areas would be expected to positively correlate with INP concentrations in any case. This is not obvious to me given the low strength of the marine source of INPs and the ready ability of terrestrial sources to dominate with any exposure to land emissions, which has been clear in some past studies. Specific questions/comments for addressing before publication are listed below. The paper is otherwise very well written, so these extra comments are limited to the topics overviewed above.

**Specific Comments**

*[1]* Page 3, lines 17-20: Could the meaning of the wind directions noted be clarified? Do these imply from open water? Or assumed free from stack contamination? This actually touches on the topic of using remote ocean data and assuming marine only influence of course, about which not much is said.

*[A1] The wind directions noted correspond to conditions assumed to be free from stack contamination. To address the referee's comment this information has been added to the revised manuscript. Specifically the following text has been added to the manuscript (Section 2.1):*

*"The data reported in DeMott et al. (2016) only included sites in Baffin Bay, days where it did not rain and conditions when the samples were assumed to have reduced exposure to ship smokestack contamination i.e. when the apparent wind direction measured on the ship was between 0-90 degrees or 270-360 degrees, where 0/360 corresponds to the bow of ship"*

*[2]* Page 5, lines 18-19: Can you justify not considering the freezing of drops not on a spot as another sort of background freezing spectrum? Is it because you cannot be certain that some small particles did not migrate under this position on the slide?

*[A2] Correct - we cannot be certain that some small particles do not migrate under this position on the slide. As stated on Page 5, Lines 19-21, "We assumed these relatively rare occurrences were due to particles < 0.18 μm in diameter that were not focused into spots or due to rebound of a small fraction of the particles off the hydrophobic glass slides". We do not consider this freezing as another sort of background freezing spectrum since no freezing from the blanks was observed at temperatures ≥-25 °C.*

*[3] Page 7, lines 3-5: This statement about the footprint layer confused me because I could not clearly distinguish how this differed from the earlier statement that the terminal point was 0 to 60 m above sea level. Please add, if possible.*

*[A3] At the start of the simulation the particles are released in a box and are followed backwards in time. For our study the box had the dimensions 100 x 100 m in the horizontal and from 0 to 60 m in the vertical. This initial box is only relevant for the release, after which the particles are free to move in all dimensions. After release of the particles, they are followed backwards in time to determine potential emission sensitivity (PES) plots. Since we are concerned with emissions from the surface we determined PES plots for the footprint layer (0 - 300 m).*

*We have added text to the manuscript to clarify that 0 - 60 m in the vertical only relates to initial simulation conditions.*

*[4] Page 7, lines 19-20: One sample with "low" INP was selected, but I see only 4 or 5 samples in Fig. 2 that even have data at -15 ∘C, so if three of these were used, then they are already not low INP samples I would judge. Is there a possible bias in the samples selected? After all, INP concentrations of 0.1 per liter at this temperature are fairly classical continental numbers in the first place. Your qualification about the conclusions being applicable only to the CCSEM cases on page 9 is duly noted. I wonder if you might comment about the influence of selection of samples for microscopy.*

*[A4] We chose to use two samples that showed high INP concentrations and one sample with a low INP concentration. The 29$^{th}$ July sample did not display freezing activity above -25 °C, which is why it was considered a low INP concentration day. We explored two days with higher INP concentrations, the 25$^{th}$ July and the 21$^{st}$ July. These two days were random choices. We have adjusted the text in section 3.2 to read:*

*"The two samples corresponding to high [INP(T)] were collected on July 21$^{st}$ and 25$^{th}$, these days were chosen randomly out of the samples that showed freezing activity at - 20 °C and -25 °C. The sample corresponding to a low [INP(T)] was collected on July 29$^{th}$, this day was chosen randomly out of the samples that did not display any freezing activity above -25 °C. "*

*[5] Page 7, lines 23-24: Please clarify if fully quantitative EDS analysis is performed to determine atomic percentages. That is the counts are interpreted quantitatively or qualitatively? Figure S4 shows actual concentration values as a means of interpreting mixtures as sea salt or dust, but there are no mixtures. It would be interesting to know how many of the dusts were salty, as processing may potentially alter their assumed behaviors (i.e., compared to parameterized dust).*

*[A5] A fully quantitative EDX analysis was performed to determine atomic percentages. For the current analysis, we combined particles classified as mineral dust and mixtures of mineral dust and sea salt together, as illustrated in Figure S4. If we used the*

*classification scheme by Derimian et al. (2017) that explicitly classifies particles as sea salt, mineral dust, and mixed sea salt/mineral dust, for the three samples analysed, no particles are classified as mixed sea salt/mineral dust. The text in the manuscript has been adjusted to the following:*

*"This classification scheme does not include a mixed mineral dust/sea salt particle category. Rather, any particles that have both sea salt and mineral dust are classified as either mineral dust or sea salt depending on the largest atomic percentage contribution. If we used the classification scheme by Derimian et al. (2017), which explicitly classifies particles as sea salt, mineral dust, and mixed sea salt/mineral dust, no particles would be classified as mixed sea salt/mineral dust particles."*

*[6]* Page 7, lines 27-28: Please also tabulate surface areas attributed to mineral dust and sea salt. It seems that one could also derive ns for each of these. This is critically important to the assumption that these can simply be applied in existing parameterizations that will then be turned into a ratio according to your Eq. 3.

*[A6] To address the referee's comment the surface areas attributed to mineral dust and sea salt have been added in Table S2 in the Supplement. Derivation of $n_s$ is discussed below.*

*[7]* Page 8, lines 18-21: Following from what I just said, if you have surface area, can't you compute ns and compare to parameterizations on the assumptions that particles are all SSA or all minerals, or use the mineral ratio to parse it out? Are the results consistent in any way with the parameterization or does the dust act differently than expected? Otherwise, by applying Niemand et al., you are making assumptions about the activity of the dust in these samples that may or may not be true. The same could be said about the marine spectrum, but the dust appears to dominate here, so is the most critical assumption.

*[A7] To address the referee's comment we have added calculations of $n_s$-values of mineral dust to the revised manuscript and we have compared the calculated $n_s$-values to data in the literature for mineral dust.*

*[8]* Page 9: The correlation analyses are not especially impressive on first glance. Nevertheless, it seems to make sense that residence time over land would have a positive impact on INP concentrations, while time over water would not be expected to matter. This is the natural result when the land surface is a so much more powerful source by the amount that has been noted in past studies. The marine source would only be expected to show up when there is little or no land influence. Both sources would seem to depend more on wind conditions.

*[A8] We agree with the referee.*

*[9]* Page 10, lines 11-12: As noted, there is field evidence for marine INP ice active surface site densities in line with what is assumed for the exercise here.

*[A9] In the revised manuscript we have referenced the field evidence for marine INP ice active surface site densities.*

*[10]* Page 10, line 14: The comment about biological INPs is kind of a hanging thought. Are you inferring that the results are influenced by these? Your references suggest a range of source types including microbes or macromolecules it seems. While one might agree with the sentiment, it is not clear how it relates to the subject of this paper and why it only appears in the conclusions.

*[A10] We have removed the comment about biological INPs.*

*[11]* Page 10, lines 26-27: While the correlation analyses and dispersion modeling support the role of mineral dust especially in case of higher loading (e.g., a few to 10% of surface area represented is a good amount of mineral dust), this does not indicate at what level marine INPs become important for lower INP concentrations does it? In a number of cases, INPs were apparently below detection limits, possibly consistent with typical surface areas and marine INP ns, as well as with limited sample volumes assessed (limited warm temperature assessment). And again, any time spent over land would be expected to influence INPs strongly, while time spent over what might be a more constant and low INP source like the ocean would not be expected to correlate with INP concentrations. Those numbers may be relatively uniform independent of time spent over the marine source, but instead more correlated with marine conditions that influence emission rates. The fractional marine influence does not seem fully evaluated, as it would have required assessing more cases at the lower end of the spectrum of measurements I suspect.

*[A11] We agree with the referee.*

*We have modified this discussion from the following:* "This correlation analysis together with the particle dispersion modelling provides further evidence, that sea spray aerosol was likely not the major source of INPs during sampling."

To:

"This correlation analysis together with the particle dispersion modelling suggests that sea spray aerosol was likely not the major source of INPs during sampling, at least not when INP concentrations were high. Sea spray aerosol may still have played a role when the INP concentrations were low during sampling."

**References**

McCluskey, C. S., Ovadnevaite, J., Rinaldi, M., Atkinson, J., Belosi, F., Ceburnis, D., et al. (2018). Marine and terrestrial organic ice-nucleating particles in pristine marine to continentally influenced Northeast Atlantic air masses. Journal of Geophysical Research: Atmospheres, 123. https://doi.org/10.1029/2017JD028033

**Anonymous Referee #2**

**Summary**

This study researched the quantity, chemistry, and potential sources of immersion mode ice nucleating particles in the Canadian Arctic marine boundary layer during the summer of 2014. Aerosol samples were collected at 28 locations on a MOSSI impactor and then analyzed using a droplet freezing technique to quantify the concentration of INPs as a function of temperature. The ratio of mineral dust to sea spray particle surface area was quantified using EDX for three samples. These surface area ratios were then converted to active site density ratios, which revealed that mineral dust was the dominant INP type in the considered samples. Finally, the authors use the FLEXPART particle dispersion model to correlate INP concentration data with air mass back trajectories. This analysis suggests the source of INPs in the arctic marine boundary layer are relatively local mineral or soil dusts.

**General Comments**

This study makes a substantial contribution to the field by presenting an impressive amount of immersion mode INP concentration data for the Arctic. The locality of the field work is very timely, and the additional analyses on particle composition and back trajectories nicely support the authors' main goal of quantifying immersion mode INP concentrations. The paper is well suited for publication in ACP. Below, I provide a few questions and comments to strengthen the paper and clarify its results.

**Note on INP Nomenclature**

*[12]* Please try to be clearer throughout the text that you considered the immersion freezing mode of ice nucleation. This is especially important given your results that mineral dust was an important source of particulate matter in your samples: mineral dust can activate in the deposition freezing mode at the temperatures considered here (e.g. Hoose and Mohler, 2012), but the mode of freezing investigated in this study isn't even included in the abstract.

*[A12] We have clarified the mode of ice nucleation to be immersion freezing throughout the manuscript.*

**Specific Comments**

*[13]* Page 2, Line 9: Please clarify here that sea spray particles actually vary widely in composition, depending on mechanism of formation. For instance, film rupture particles are organic enriched compared to jet droplet particles (e.g. Wang et al. 2017). This will highlight the importance of considering both microlayer and bulk seawater samples, as you suggest in the following sentence.

*[A13] We have clarified that sea spray particles vary widely in composition depending on mechanism of formation. This has been done by amending the text to:*

*"Sea spray aerosol is generated by a bubble bursting mechanism at the ocean surface (Blanchard, 1964) and varies widely in composition, depending on the mechanism of formation. For example, particles from film rupture are enriched in organics compared to particles from jet droplets (Wang et al., 2017)."*

*[14]* Page 2, Line 11: For completeness, the authors may also consider citing Huang et al. 2018 here. This paper follows up on Burrows et al. 2013 by discussing the sensitivity of the relative contribution of sea spray to INP concentrations at high latitudes to a model's choice of active site density parameterizations.

*[A14] We have added the citation Huang et al. (2018) to the manuscript.*

*[15]* Page 3, Line 15: Were the metrological data collected with on-ship sensors? If so, can some estimate of the height of these sensors above sea level be given?

*[A15] The meteorological data was collected with on-ship sensors. We have added the height of the sensors used to measure RH, wind speed, and temperature to the manuscript.*

*[16]* Page 4, Line 8: Has the transmission efficiency of the louvered total suspended particulate (TSP) inlet been quantified before? For example, it sounds similar to an inlet characterized by Kenny et al. 2005. If the exact loss percentages have not been measured, that's okay; but the authors should note that larger particles (like mineral dust) may be more prone to impaction and thus be undersampled on the MOSSI slides.

*[A16] The transmission efficiency of the louvered total suspended particulate inlet has been quantified previously (Kenny et al., 2005). Based on this previous study, the transmission efficiency of 10 $\mu$m particles through the inlet is $\geq$ 90 %. To address the referee's comments, this information has been added to the revised manuscript.*

*[17]* Page 6, Line 7: Was the height of the chemical ionization mass spectrometer inlet at a similar height to the inlet of the MOSSI?

*[A17] We have added the height of the CIMS to the manuscript.*

*[18]* Page 7, Line 17: The authors state that only particles at the edge of the spots were analyzed with EDX, but how are they sure these particles are representative of the bulk of particles directly below the micro-orifices? For example, the authors mention the particle rebound effect (Page 4, Line 12). Can the authors discuss if perhaps mineral dust is more prone to rebound than the deliquesced salty particles, and therefore may be more likely to end up at the edge of the spot where EDX was performed?

*[A18] The particles on the edge of the spots were directly under the nozzles, meaning they were part of the spot. We assume the composition of the edge of the spot is the same as the whole spot; however, we are unable to confirm this assumption. In the revised manuscript we have clearly stated this assumption by adding the following:*

"Since the edge was still directly under the nozzle, we assumed the composition of the edge of the spot is the same as the composition of the whole spot, although we are unable to confirm this assumption."

*[19] Page 7, Line 22: "First, the atomic percentages of each particle were determined from EDX spectra." Can the authors clarify if only one discrete spot of each particle was examined, or if the EDX data represents an average signal from a raster scan?*

*[A19] To address the referee's comment we have clarified this point in the manuscript by adding the following sentence:*

"The EDX data for an individual particle represents an average signal from a raster scan."

*[20] Page 7, Line 23: Do the authors have any way to estimate what fraction of particles were internally mixed; i.e. mineral dust coated with sea salt?*

*[A20] See [A5] above.*

*[21] Page 8, Line 4: How were blanks prepared and treated? Were they fresh slides out of the package? Or were they treated the same way as the sampled slides, i.e. placed within the MOSSI only without turning on the pump? The latter would better account for contamination in the sample handling and preparation process; but in either case, more information is needed here.*

*[A21] We have included more information on how the blanks were prepared and treated in the manuscript.*

*[22] Page 8, Line 17: "The two samples corresponding to high [INP(T)] were collected on July 21st and 25th." As I read Figure 2, these are among the highest but not the highest INP concentration days. Can the authors provide any more reasoning behind why these particular days were chosen for compositional analysis? How typical was the meteorology on these days? Or perhaps they were chosen to be evenly spaced in time and location? Or was the choice random?*

*[A22] We analysed two days with higher INP concentrations, the 25$^{th}$ July and the 21$^{st}$ July. These two days were random choices. We have adjusted the text in section 3.2 to read:*

"The two samples corresponding to high [INP(T)] were collected on July 21$^{st}$ and 25$^{th}$, these days were chosen randomly out of the samples that showed freezing activity at -

20 °C and -25 °C . The sample corresponding to a low [INP(T)] was collected on July 29[th], this day was chosen randomly out of the samples that did not display any freezing activity above -25 °C."

*The 21[st] and 25[th] had 5 m/s and 3 m/s average wind speeds, respectively. The average wind speed during sampling was 5 m/s. The samples were not chosen because of the meteorology.*

*[23]* Page 10, Line 14: The jump to biological particles is a bit of a non-sequitur. You might elaborate on how biological activity can influence sea spray particle composition, which in turn effects INP activity.

*[A23] We have taken this statement out of the manuscript.*

*[24]* Page 10, Conclusions: Can the authors discuss their findings that mineral dust immersion INPs dominated over sea spray INPs in the context of Irish et al. 2017, which found immersion mode INPs to be abundant in the seawater from this region? E.g. perhaps the INPs were present in the seawater but were never aerosolized because it wasn't windy enough to generated sea spray. This also ties in with Reviewer 1's comment that time spent over open ocean may not correlate with sea spray INP concentration if sea spray particles are not being formed. Then, include a discussion on whether these conditions are typical: how do your wind speeds compare with average (intra- and interannual) wind speeds in the region? Briefly delving into a reanalysis dataset (e.g. ECMWF ERA-Interim) might help you to explore this question in depth, or you could look for historical observational data. This will require a little bit more work on your end, but it would strengthen your analysis immensely by allowing you to hypothesize whether your findings are typical.

*[A24] We have addressed the referee's comment by adding the following discussion in the manuscript:*

"Previous studies have shown that INPs are ubiquitous in Arctic seawater (Irish et al., 2017; Wilson et al., 2015). Our results show that INPs in the Arctic seawater are not emitted at a high enough rate to compete with mineral dust, at least for the samples with high INP concentrations. The flux of sea spray aerosol to the atmosphere is a function of the wind speed. The average minute wind speed during sampling (5.5 m s$^{-1}$ with 10[th] and 90[th] percentiles of 2.1 and 10.0 m s$^{-1}$) was similar to the average minute wind speed during the whole campaign (5.2 m s$^{-1}$ with 10[th] and 90[th] percentiles of 1.5 and 9.3 m s$^{-1}$). In addition, our average wind speeds were similar to historical monthly average wind speed data from Alert, NU, Canada (3.6 m s$^{-1}$ in July and 3.3 m s$^{-1}$ in August; climate.weather.gc.ca; climate ID: 2400300). The influence of sea spray aerosol may be more important during periods of higher wind speeds. On the other hand, high wind speeds are also likely to increase the flux of mineral dust from local sources."

*[25]* Page 20, Table 1: Please include in the table caption the length of the back-trajectory (7 days).

*[A25] The length of the back-trajectory has been included in the caption of Table 1.*

*[26]* Page 21, Figure 1: Can you also denote in this figure where the samples for chemical/EDX analysis were collected? That will help us visualize the potential sources of these particles in light of the data in Figure 5. Otherwise, this figure looks great!

*[A26] We have indicated which sampling locations correspond to samples used for CCSEM-EDX analysis.*

*[27]* Page 23, Figure 3: Please also show the min/max INP concentration at each temperature (in addition to the average and standard deviation you've already shown) to provide more information on the degree of variability. Are the data from DeMott et al. 2015 averages, or discrete measurements?

*[A27] We have clarified that the data from DeMott et al. (2016) are discrete measurements. The min/max INP concentrations at each temperature have also been added to the figure.*

*[28]* Page 25, Figure 5: Please increase the font size in this figure to something legible.

*[A28] The font size in Figure 5 has been increased.*

*[29]* Supplement, Page 9, Figure S6: Please include units ($L^{-1}$) for INP concentrations either in the axis label or the figure caption.

*[A29] ($L^{-1}$) has been added to the axis label in Figure S6 (now Figure S7).*

5   mixed sea salt/mineral dust, no particles would be classified as mixed sea salt/mineral dust particles.

After each particle was classified, the surface areas of mineral dust particles and sea salt particles were determined using 2D projected images recorded by SEM. Note that the actual surface area of mineral dust is underestimated using this method due to surface irregularities and complex topology. The ratio of mineral dust surface area to sea salt surface area was then determined by dividing the surface area of mineral dust for each sample by the surface area of sea salt for the same
10   sample.

**3 Results and discussion**

**3.1 Measured INP concentrations**

The measured concentrations of INPs in the immersion mode, *[INP(T)]*, sampled in the Arctic are shown in Figs. 2b and 2c. The measured *[INP(T)]* on new hydrophobic glass slides taken straight from the package, cleaned with ultra pure
15   water, but not exposed to atmospheric particles (referred to as "blanks") are shown in red in Figs. 2a and 2c. Freezing of the blanks occurred over the range of -25.9 °C to -38.4 °C. For the droplet sizes and cooling rates used here homogeneous freezing occurs at approximately -37 °C. Therefore the freezing that occurred in the blanks at temperatures above approximately -37 °C was due to heterogeneous freezing likely caused by the hydrophobic glass slides. In the following we will focus on *[INP(T)]* at temperatures of -25 °C and warmer since no freezing from the blanks was observed in this
20   temperature range. A full time series of *[INP(T)]* at -15 °C, -20 °C, and -25 °C are reported in Fig S5.

In Fig. 3 we compare recent measurements of *[INP(T)]* from several field campaigns in marine environments with the average concentrations measured in the current study. Figure 3 illustrates that the average INP concentrations measured in the current study fall within the range of INP concentrations measured in other marine environments. This observation, however, does not confirm that sea spray aerosol was the major source of INPs during the studies reported here.

25   ### 3.2 Measured ratios of mineral dust surface area to sea salt surface area

For three samples (two with high *[INP(T)]* and one with low *[INP(T)]*), we calculated the ratios of mineral dust surface area to sea salt surface area using CCSEM-EDX. The two samples corresponding to high *[INP(T)]* were collected on July 21st and 25th, these days were chosen randomly out of the samples that showed freezing activity at -20 °C and -25 °C. The sample corresponding to a low *[INP(T)]* was collected on July 29th, this day was chosen randomly out of the samples
30   that did not display any freezing activity above -25 °C. In Table S2 we report the total number of particles analysed by CCSEM-EDX for each sample, the fraction of particles classified as mineral dust and sea salt particles, and the surface area

Victoria Irish 2018-12-17 11:17 AM
**Comment [13]:** Addresses referee 1 [5] and referee 2 [20]

Victoria Irish 2018-11-19 1:35 PM
**Comment [14]:** Addresses referee 2 [12]

Victoria Irish 2018-12-17 11:25 AM
**Comment [15]:** Addresses referee 2 [21]

Victoria Irish 2018-12-17 11:11 AM
**Comment [16]:** Addresses referee 1 [4] and referee 1 [22]

corresponding to mineral dust and sea salt particles. Shown in Fig. 4a are the calculated ratios of mineral dust surface area to sea salt surface area using surface area measurements from CCSEM-EDX. For the three samples, this ratio ranged from 0.03 to 0.09. Using this ratio we estimated the ratio of $[INP(T)]$ from mineral dust, $[INP(T)]_{MD}$, to $[INP(T)]$ from sea spray, $[INP(T)]_{SS}$, using the following equation:

$$\qquad \frac{[INP(T)]_{MD}}{[INP(T)]_{SS}} = \frac{n_s(MD).S_{MD}}{n_s(SS).S_{SS}} \qquad\qquad (3)$$

Where $n_s(SS)$ is the ice active surface site density for sea spray aerosol, $n_s(MD)$ is the ice active surface site density for mineral dust, and $S_{SS}$ and $S_{MD}$ are the total surface areas measured by CCSEM-EDX for sea salt and mineral dust, respectively. The $n_s(SS)$-values were determined using laboratory data from DeMott et al. (2016). Recent studies show that $n_s(SS)$-values determined from DeMott et al. (2016) are consistent with values determined in pristine marine environments

10 (McCluskey et al., 2018). The $n_s(MD)$-values were calculated using the exponential function reported by Niemand et al. (2012) that was determined from freezing data of Asian dust, Saharan dust, Canary Island dust, and Israel dust. For details see Section S1.

The ratios of INP concentrations based on Eq. 3 for freezing temperatures of -25, -20, and -15 °C, are shown in Figs. 4b, 4c, and 4d respectively. These ratios suggest that for the three samples when CCSEM-EDX measurements were

15 performed, the $[INP(T)]_{MD}$ are higher than the $[INP(T)]_{SS}$ (ratios were between 10 to $10^3$, inclusive of errors, at -15, -20 and -25 °C), assuming the $n_s$-values used are applicable for the field studies reported here. These results also suggest that mineral dust is a more important contributor to the INP population than sea spray aerosol for the times and locations corresponding to the CCSEM-EDX measurements.

Above we assumed that the ice active surface site density for mineral dust, $n_s(MD)$, in our studies can be calculated

20 with the exponential function reported by Niemand et al. (2012) that was determined from freezing data of Asian dust, Saharan dust, Canary Island dust, and Israel dust. To test this assumption, we calculated $n_s(MD)$-values using the CCSEM-EDX measurements on July 21st and 25th. Details on how $n_s(MD)$-values were calculated is given in Section S2, and the results are shown in Fig. S6. In short, within the uncertainty of the measurements, our calculated $n_s(MD)$-values are consistent with the results from Niemand et al. (2012).

25 **3.3 Particle dispersion modelling**

Figure 5a shows the averaged PES for the footprint layer for all samples combined and suggests that the source of INPs sampled during the campaign was local (i.e. the Canadian Arctic Archipelago, Baffin Bay, and eastern Greenland).

Figure 5b shows the averaged PES for the footprint layer for samples that had the highest INP concentrations (top 36 % of the samples) at -25 °C. Figure 5c shows the average PES for the footprint layer for samples that had the lowest INP

30 concentrations (bottom 36 % of the samples) at -25 °C. A cut-off of 36 % was selected since no freezing was observed in 36 % of the samples at -25 °C. Figure 5d shows the ratio of the average PES for the highest INP concentrations to the average

Victoria Irish 2018-12-17 11:28 AM
Comment [17]: Addresses referee 1 [6]

Victoria Irish 2018-12-17 11:30 AM
Comment [18]: Addresses referee 1 [9]

Victoria Irish 2018-12-17 1:09 PM
Comment [19]: Addresses referee 1 [7]

[revised manuscript text omitted]

Victoria Irish 2018-12-17 1:18 PM
**Comment [20]:** We have removed the comment about biological INPs. Addresses referee 1 [10] and referee 2 [23]

Victoria Irish 2018-12-17 12:58 PM
**Comment [21]:** Addresses referee 1 [11]

Victoria Irish 2018-12-17 1:02 PM
**Comment [22]:** Addresses referee 2 [24]

[revised manuscript text omitted]